# Early establishment and life course stability of sex biases in the human brain transcriptome

## Graphical abstract

## Authors

Clara Benoit-Pilven, Juho V. Asteljoki, Jaakko T. Leinonen, Juha Karjalainen, Mark J. Daly, Taru Tukiainen

## Correspondence

taru.tukiainen@helsinki.fi

## In brief

Benoit-Pilven et al. explore the origins of the well-established sex differences in many brain-related phenotypes. Through comparing sex-biased gene expression in the prenatal and adult human forebrain, they show early development is a crucial time point for the introduction of sex differences and propose that both hormones and sex chromosomes shape the brain sex biases.

## Highlights

- Sex differences in gene expression (sex-DEs) are abundant in early brain development

- A large fraction of the sex-DEs in the developing human brain persists until adulthood

- Both sex-chromosomal and hormonal factors contribute to the brain sex-DEs

- Sex-DE genes show no overlap but rather interact with disease-associated genes

Benoit-Pilven et al., 2025, Cell Genomics 5, 100890
July 9, 2025 © 2025 The Author(s). Published by Elsevier Inc.

CellPress

# Early establishment and life course stability of sex biases in the human brain transcriptome

Clara Benoit-Pilven,[1] Juho V. Asteljoki,[2,3,4] Jaakko T. Leinonen,[1] Juha Karjalainen,[1,5] Mark J. Daly,[1,5,6,7] and Taru Tukiainen[1,8,*]

[1]Institute for Molecular Medicine Finland FIMM, HiLIFE, University of Helsinki, Helsinki, Finland
[2]Minerva Foundation Institute for Medical Research, Helsinki, Finland
[3]Department of Internal Medicine, University of Helsinki, Helsinki, Finland
[4]Abdominal Center, Helsinki University Hospital, Helsinki, Finland
[5]Program in Medical and Population Genetics, Broad Institute of Harvard and MIT, Cambridge, MA, USA
[6]Stanley Center for Psychiatric Research, Broad Institute of Harvard and MIT, Cambridge, MA, USA
[7]Analytic and Translational Genetics Unit, Massachusetts General Hospital, Boston, MA, USA
[8]Lead contact
*Correspondence: taru.tukiainen@helsinki.fi

## SUMMARY

To elaborate on the origins of the established male-female differences in several brain-related phenotypes, we assessed the patterns of transcriptomic sex biases in the developing and adult human forebrain. We find an abundance of sex differences in expression (sex-DEs) in the prenatal brain, driven by both hormonal and sex-chromosomal factors, and considerable consistency in the sex effects between the developing and adult brain, with little sex-DE exclusive to the adult forebrain. Sex-DE was not enriched in genes associated with brain disorders, consistent with systematic differences in the characteristics of these genes (e.g., constraint). Yet, the genes with persistent sex-DE across the lifespan were overrepresented in disease gene co-regulation networks, pointing to their potential to mediate sex biases in brain phenotypes. Altogether, our work highlights prenatal development as a crucial time point for the establishment of brain sex differences.

## INTRODUCTION

A variety of human morphological, physiological, and behavioral phenotypes differ between male and female individuals. For instance, sex differences exist in the structural and functional brain organization.[1–3] Also, numerous neurological and psychiatric disorders display well-established differences in incidence, prevalence, severity, and/or age at onset between sexes.[4] For example, autism spectrum disorder (ASD) shows clear sex biases in prevalence (4.5:1 male-to-female ratio) and severity (more female ASD patients presenting with intellectual disability).[5,6] Despite these abundant sex differences related to the human brain function, little is known about how and when they appear.

Although genetic factors often explain a large fraction of the phenotypic variability, there is limited evidence that genetic architecture has a substantial contribution to sex differences in complex traits.[7,8] For brain-related traits and diseases, genetic studies have found only minor sex differences both in common variant associations[1] and in rare variant burden,[9,10] although a difference in the liability threshold between sexes may play a part in the sex-biased prevalence of ASD.[11,12] As genetic studies have generally been unsuccessful in explaining the phenotypic sex differences, studies have searched for alternative explanations at other levels of regulation, i.e., in epigenetics[13,14] and transcriptomics.[15]

In contrast to the limited sex differences in the genetic regulation of phenotypes,[8,15] a growing number of transcriptomic studies point to the abundance of sex-differentially expressed (sex-DE) genes in the adult brain.[15–19] However, as gene expression throughout the lifespan is dynamic,[20] transcriptomic sex differences in the adult brain may be distinct from the processes leading to phenotypic sex biases, as many neurodevelopmental disorders, including ASD and attention-deficit/hyperactivity disorder (ADHD), are rooted in fetal development.[21,22] Gene expression sex biases in the prenatal brain have also been described,[16,17,23–26] yet most studies have been limited by the small sample sizes and the heterogeneity of the data, including large variation of gestational ages and brain regions sampled. One of the largest studies thus far[24] analyzed 120 prenatal whole brain samples from the second trimester of pregnancy, discovering more than 2,000 differentially expressed genes between males and females. Such a large number of sex-DE genes uncovered suggests sex biases in brain gene expression appear very early during fetal development.

The brain tissue is complex with specific regions displaying distinct characteristics, functions, and transcriptomic patterns of expression.[27,28] The forebrain is one of the three primary vesicles of the brain during development that can be identified as early as 3–4 weeks post-conception (PCW) in humans.[29] This vesicle gives rise to the cerebral cortex, which composes around two-thirds of the adult brain's total mass, as well as to the

amygdala, the hippocampus, the hypothalamus, and the basal ganglia. The forebrain is implicated in the higher brain functions, such as thinking, memory, and sensory processing, and is affected by neuroanatomical alterations in several neurological diseases, including ASD[30,31] or schizophrenia,[32] proposing this brain region as a particularly interesting target for the study of sex differences.

In this study, we explored the extent, dynamics, and biology of transcriptomic sex differences in the developing and adult human forebrain to provide insights into the timing and mechanisms of the widespread brain-related male-female differences. To this end, we used RNA sequencing (RNA-seq) datasets from the early prenatal, i.e., first and second trimester, and adult forebrain totaling together 1,899 samples. We assessed the patterns of sex-DEs in and between prenatal and adult forebrain and investigated the potential causes and consequences of these transcriptomic sex differences. Overall, our work highlights abundant functionally relevant male-female differences in the human forebrain emerging in early development that can plausibly contribute to the sex differences detected in diverse brain-related phenotypes.

## RESULTS

### Description of datasets

We analyzed the gene expression signatures in two forebrain datasets, one from the prenatal brain and the other from the adult brain. For the prenatal dataset, we extracted forebrain samples from the Human Developmental Biology Resource (HDBR) Expression dataset[33] (see STAR Methods) resulting in 266 RNA-seq samples (130 samples from 37 female individuals and 136 samples from 35 male individuals, with a median of two samples per individual) from the first and early second trimester (5–17 weeks post-conception [PCW]) pregnancies (Figures 1A and S1A; Table S1). All analyzed HDBR samples were collected from voluntary interruptions of pregnancy and were previously confirmed to be karyotypically normal.[33,34] The adult brain samples were obtained from the Genotype-Tissue Expression project (GTEx v8).[35] We selected samples from the forebrain region (see STAR Methods), accounting for a total of 1,633 RNA-seq samples (434 samples from 91 female individuals and 1,199 samples from 246 male individuals, median of five samples per individual) (Figure S1B; Table S1). In addition to its biological relevance to many neurodevelopmental diseases,[36] we focused on the forebrain region to ensure a dataset as homogeneous as possible (see STAR Methods; Figure S2). Our definition of male and female individuals refers to genetic sex (i.e., XY and XX), determined via sex chromosome gene expression (see STAR Methods).

### Pseudotime inference for prenatal forebrain samples

The age range of the HDBR samples (Figure S1A) covers a highly active developmental period,[29] resulting in transcriptional heterogeneity of the data (Figure S2D). Using a cell-type decomposition analysis with CIBERSORTx,[37] we confirmed this period of development coincides with the emergence of neurons (mean proportion of neurons ranging from 10% to 73% in the first

[5–7PCW] and the last [14–17PCW] developmental group; Figures S3A and S3B). Given these differences, we chose to account for this developmental trajectory in the joint analysis of the prenatal samples. The classification of the samples into discrete post-conception-week categories, however, does not fully reflect the reality of development, which is a continuous and complex process.

Given the successful use of pseudotime inference in single-cell studies to model time course experiments,[38] we reasoned a similar approach could be applied to estimate a trajectory of early brain development in the HDBR samples. We therefore estimated a continuous pseudotime variable from the gene expression data using the phenopath R package[39] (see STAR Methods; Figures S3C and S4A). The inferred pseudotime variable correlated well with the reported developmental stages (Kendall's tau = 0.46; Figure 1B), suggesting it provides a continuous ordering of the samples that, at least partly, reflects the progression of brain development.

We performed several additional analyses (see STAR Methods, Figures S4–S6) to confirm the robustness of the pseudotime approach. First, we identified a clear correlation between cell type composition (obtained with CIBERSORTx; see STAR Methods) and pseudotime (Figure S4D) in line with pseudotime tracking biological processes associated with the developmental progression. This trend was also captured using a different pseudotime inference method, monocle2 (Figures S4E and S4F). We noted the inferred pseudotime remained similar irrespective of the number of input genes to the phenopath algorithm (between 1,000 and 2,000 most variable genes) and identified a set of 100 genes most relevant for the correlation with the reported developmental stages that together point toward processes related to development and neurogenesis (Figure S5). Finally, we found that including developmental stage as a covariate in the pseudotime inference expectedly reduced the correlation with the reported developmental stages (Kendall's tau = 0.17). This result further supports the idea that pseudotime analysis in HDBR can, indeed, find patterns in the expression data that in part reflect early brain development.

Demonstrating pseudotime inference is largely unaffected by the HDBR data containing multiple samples per individual (mean number of samples per individual = 3.7, originating from different regions of the forebrain), a pseudotime inferred including only one sample per individual provided a consistent ordering of the individuals in the study set to the original pseudotime (Pearson's r = 0.82) and was highly correlated with the developmental stages (Kendall's tau = 0.67). However, the correlation with cell type composition was less pronounced with the pseudotime inferred from only one sample per individual, which suggests the original pseudotime partly captures also forebrain region-specific expression differences within a developmental stage (Figure S6). We additionally confirmed that the inferred pseudotime was more consistent between samples from the same individual than between random samples (Figure S4B). Overall, these findings show the inferred pseudotime variable reflects meaningful biology and allows ordering the samples along the development in a continuous manner.

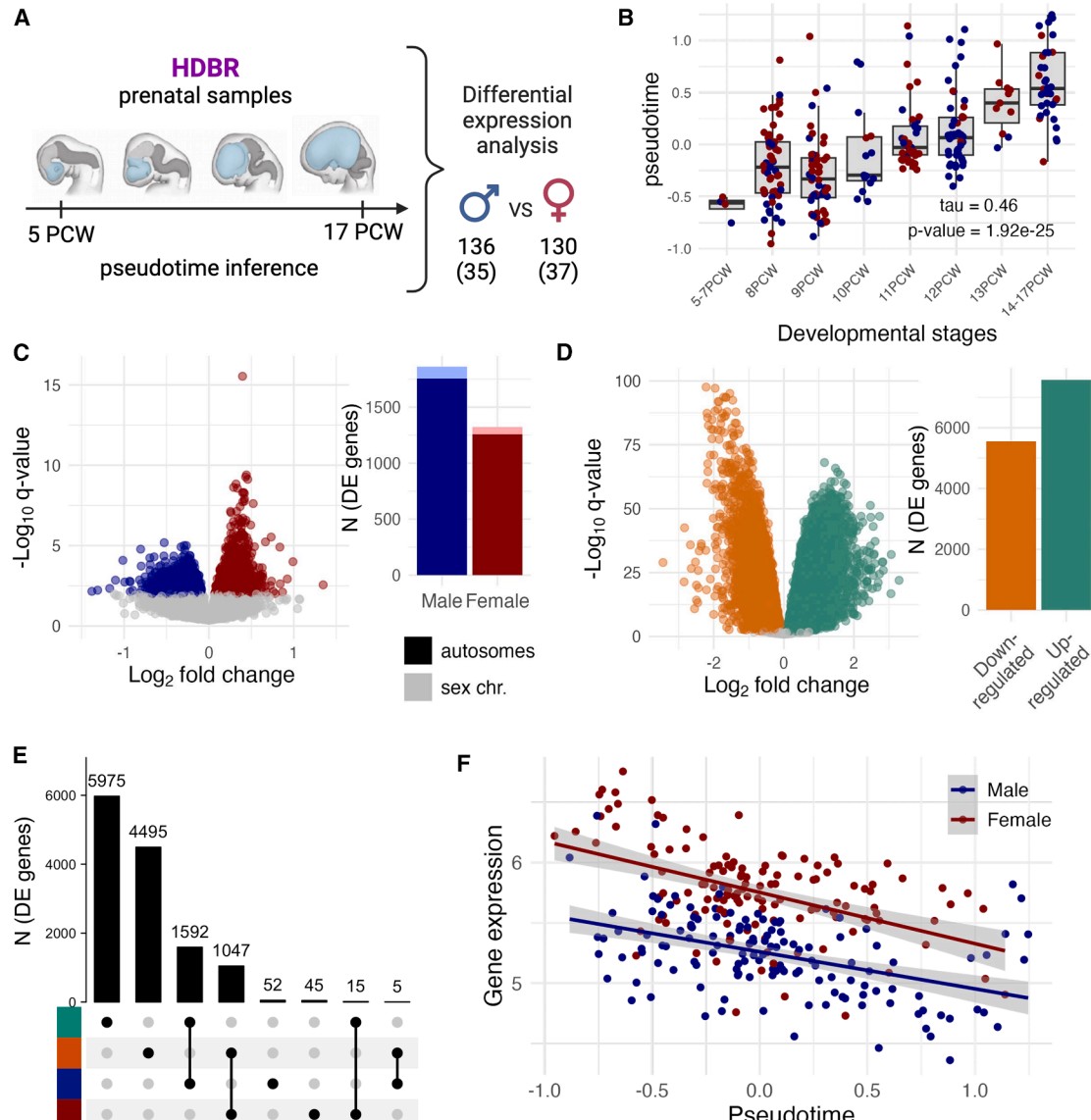

**Figure 1. Differential expression analysis in the prenatal forebrain**

(A) The prenatal dataset consisted of 266 samples from 72 distinct individuals from the Human Developmental Biology Resource (HDBR) ranging from 5 to 17 post-conception weeks (PCW). We first carried out a pseudotime analysis to place the samples on a continuous developmental trajectory and then performed a differential expression (DE) analysis between the male and female samples to quantify sex-differential gene expression.

(B) Correlation between the inferred pseudotime variable and the reported developmental stages in PCWs. Male samples are colored in blue, female samples in red. Kendall's tau rank correlation and its *p* value are given in the plot.

(C and D) Barplot of the number of DE genes (q value < 0.01) and volcano plot for the two analyses: (C) sex-DE genes and (D) pseudotime-DE genes. On the sex-DE volcano plot, only autosomal genes are shown. Only significant genes are colored in both volcano plots.

(E) Intersection of the findings from the two DE analyses as an upset plot taking into account the direction of effect. The upper barplot shows the size of the intersections.

(F) Expression of *ZFX* along pseudotime. *ZFX* is female-biased (q value = $4.5 \times 10^{-12}$, logFC = 0.52), and downregulated along pseudotime (q value = $2.4 \times 10^{-18}$, logFC = −0.43), but does not show any significant interaction (q value = 0.42, logFC = 0.02).

### Widespread sex biases in the prenatal brain

We carried out a sex-DE analysis across the 266 prenatal RNA-seq samples adjusting for pseudotime using a linear mixed model implemented in the variancePartition R package[40,41] (see STAR Methods; Figure S7). We found 3,187 sex-DE genes

(18.1% of all genes; 94.5% autosomal; q value <0.01) (Table S2), with a slightly higher proportion of male-biased genes (*n* = 1,864, 58.5% of all sex-DE genes, binomial test *p* value = $8.96 \times 10^{-22}$) (Figure 1C), observation in line with previous analyses of the developing brain.[16,24] Only a few of the sex-DE genes

(six autosomal and seven sex-chromosomal) showed substantial (|logFC|>1) effect sizes (Figure 1C; Table S2) with the average magnitude of differential expression between male and female samples being ~20% (mean |logFC| = 0.28). Of note, the sex-DE genes were on average more highly expressed and less constrained than the non-sex-DE ones (Figure S8).

For validation, we assessed sex-DE replacing pseudotime with the reported categorical developmental stages. All sex-DE genes (n = 31) from this analysis were also significant in the original analysis, and the high correlation of the sex-DE effect sizes (logFC) (Pearson's r = 0.838 across all significant genes in the original analysis, $p < 1 \times 10^{-10}$) further indicated these two approaches capture the same sex-DE signal (Figure S9) albeit with the continuous pseudotime covariate providing a more powered approach. Confirming the detected sex effects are generalizable across datasets, we further noted a large overlap of sex-DE genes with previous prenatal brain sex-DE analyses from other datasets[26] (87.18% and 68.66% of the significant sex-DE genes from the BrainVar and UCLA datasets, respectively, were also significant in our analysis).

In a Gene Ontology (GO) term enrichment analysis (see STAR Methods) the female-biased genes were enriched in terms associated with the cell cycle (144 of 184 significant GO terms; Table S3), potentially related to neurogenesis,[42] while male-biased genes showed enrichment in biological processes related to mitochondrial metabolism and autophagy (39 of 115 significant GO terms; Table S3) shown to be important in neuronal development[43,44] and in GO terms linked to synapses and neurons (3 of 115 significant GO terms).

### Systematic sex differences in the brain developmental trajectory

To complement the sex-DE analyses, we assessed differential expression for the pseudotime variable, which to a degree proxies the developmental progression. Illustrating the dynamics of gene expression during early brain development, many genes (13,129, 84.1%, after excluding the 3,000 genes used to infer the pseudotime) were significantly DE (q value < 0.01) along pseudotime (referred to as pseudotime-DE genes) (Figure 1D; Table S2).

A joint examination of the two DE results revealed an enrichment in the overlap between the sex-DE and pseudotime-DE genes (2,659 of 2,756 genes, 96.5%; hypergeometric test, $p = 1.4 \times 10^{-39}$ for enrichment) (Figure 1E), with considerable consistency in effect directions. Most male-biased genes were upregulated along development, i.e., more highly expressed in later pseudotime (1,592 of 1,649 genes, 96.5%, $p < 1 \times 10^{-100}$), while most female-biased genes were downregulated with pseudotime (1,047 of 1,107 genes, 94.6%, $p < 1 \times 10^{-100}$) (Figure 1E). This observation was not explained by sex differences in the mean or range in pseudotime (mean male = 0.06 [$\sigma^2 = 0.25$], mean female = −0.02 [$\sigma^2 = 0.18$]; Wilcoxon test $p = 0.26$; $F$-test $p = 0.09$), neither was it explained by a bias introduced at the pseudotime inference step, as the pseudotime-DE genes remained highly similar when a using a permuted sex variable as a covariate in the pseudotime inference (on average, 92.02% overlap in significant pseudotime-DE genes, mean Pearson's r = 0.95 of the effect sizes).

Following the large overlap of these two gene sets, similar GO terms enriched in sex-DE genes were significantly enriched among the pseudotime-DE genes (Table S3). We detected only a few genes (496, 3.2%; Table S2) with a significant interaction (q value < 0.01) between sex and pseudotime suggesting that for most genes the magnitude of gene expression sex difference remains on average constant along pseudotime (e.g., *ZFX* in Figure 1F) irrespective of the cell type compositional changes occurring in this window (Figure S3C). This agrees with a previous study reporting consistent age effects on gene expression in both sexes in the developing brain.[26]

### Attenuated sex bias in the adult brain

To understand the lifespan dynamics of the transcriptomic sex biases in the human brain, we additionally assessed differential gene expression in 1,633 adult (20–79 years) forebrain samples from the GTEx project[35] (Figure S10). Following the analysis of the prenatal forebrain, we first estimated pseudotime for each adult sample (Figure S10A). The pseudotime in adult forebrain did not correlate with the age of the samples to a similarly high degree as in the prenatal forebrain (Kendall's tau = 0.12, Figure S10B), nor did pseudotime explain as large a fraction of the variability in the data (mean variance explained by pseudotime in the prenatal forebrain = 16.4%; in the adult forebrain = 3.4%; Figures S7 and S11). These observations can point to both larger environmental variability in the adult samples and pseudotime reflecting also non-age-related expression variability in GTEx, such as the forebrain cytoarchitecture (Figure S12) and sample-specific technical factors (e.g., RNA integrity number, Pearson's correlation with pseudotime = −0.46). We, however, used pseudotime as a covariate to harmonize the GTEx analysis with the previous prenatal analysis.

We detected 1,033 sex-DE genes (6.0% of all genes; 91.4% autosomal) in the adult forebrain (q value < 0.01) (Figure S10C; Table S4), a considerably smaller number than prenatal forebrain despite the larger sample size of GTEx. Reflecting the lower number of sex-DE genes, the effect sizes of these genes in the adult brain were globally smaller than in the prenatal brain (sex-biased genes mean |logFC| 0.14 vs. 0.28). Similarly to the prenatal brain, the male-biased genes were enriched in brain-related GO terms (e.g., regulation of neurotransmitters and synaptic organization) (Table S3).

We identified 4,549 (29.7%) pseudotime-DE genes in GTEx (Figure S10D; Table S4). These genes were implicated in diverse processes such as synapse organization and autophagy for downregulated and upregulated genes, respectively (Table S3). The overlap of the sex-DE genes with the pseudotime-DE genes was significant (434 of 926 genes, 46.9%; hypergeometric test, $p = 1.63 \times 10^{-29}$ for enrichment; Figure S10E) with non-random sharing of effect directions. As opposed to the prenatal brain, in GTEx the male-biased genes were preferentially downregulated along pseudotime (283 of 610, $p = 2.0 \times 10^{-104}$), and the female-biased genes enriched in the upregulated ones (104 of 316, $p = 3.1 \times 10^{-11}$) (Figure S10E). Similarly to the prenatal brain, only a small number of genes (278, 1.8%) had a significant interaction (q value < 0.01) between sex and pseudotime (Table S4).

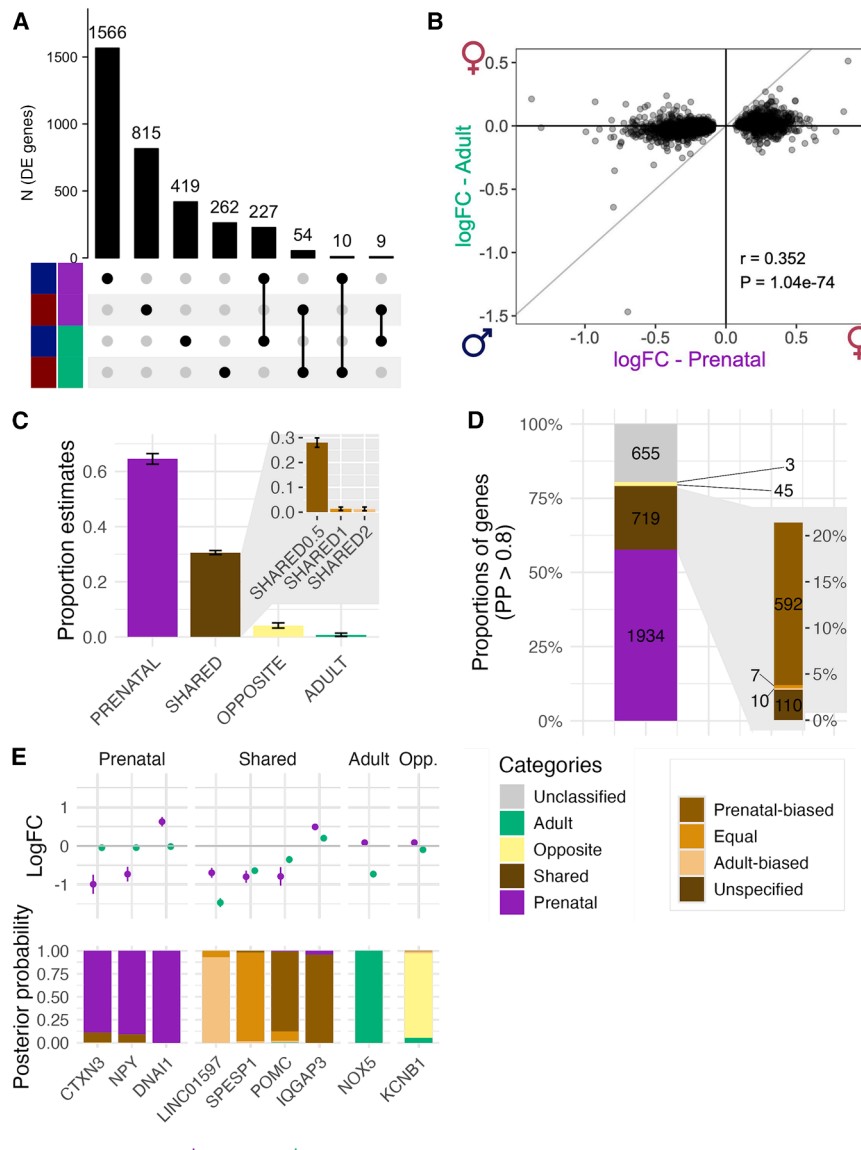

Figure 2. Comparison of the detected sex-DE genes in the prenatal (HDBR) and the adult (GTEx) forebrain

(A) Intersection of the sex-DE genes in the two datasets separated by the direction of effect (male or female-biased) as an upset plot. The upper barplot shows the size of the intersections.

(B) Scatterplot of female/male effect size (logFC) in prenatal and adult dataset for all autosomal prenatal sex-DE genes. Pearson's correlation (r) of the effect sizes and its *p* value (P) are given in the plot.

(C) Proportion estimates and SE of each gene category in the Bayesian analysis: prenatal-specific (PRENATAL), shared (SHARED), opposite effect size (OPPOSITE), and adult-specific (ADULT). The zoom plot on the top right shows the proportion and SE of the different categories of shared effect: shared with an effect size twice larger in prenatal data (SHARED0.5), shared with the same effect size (SHARED1), and shared with an effect size twice larger in adult brain (SHARED2).

(D) Proportions and numbers of individual sex-DE genes by effect category (Prenatal-specific, Adult-specific, Shared, Opposite, and Unclassified). A gene was assigned into a category if the posterior probability for a given category reached >0.8. A zoom of the different shared categories (prenatal-biased, equal, adult-biased, and unspecified) is shown on the right.

(E) Examples of autosomal genes for the prenatal-specific (*n* = 4), the shared (*n* = 4), the adult-specific (*n* = 1), and the opposite (*n* = 1) categories. The upper plot represents the logFC (with standard error) in both datasets. The lower plot shows the posterior probabilities from the Bayesian model for each gene.

## Considerable sharing of brain sex bias between early development and adulthood

As gene regulatory mechanisms are known to change across development and life stages,[45] we sought to understand the degree of sex-DE sharing between adult and prenatal brain (Figure 2; Table S5). We noted that the sex-DEs between these two life stages are widely similar (72.2% consistency of effect direction, Pearson's r = 0.60, $\pi_1$ = 0.46; Table S5; Figures 2A and 2B), irrespective of the large cell type compositional differences between the datasets (Figures S3 and S13). The similarities were still a degree lower than when comparing two independent datasets of the same tissue and age range (95.7% consistency of effect direction, Pearson's r = 0.99, $\pi_1$ = 0.87; Table S5; Figure S14), suggesting also a degree of distinct sex effects between the two life stages.

To further quantify the life-stage specificity of sex-DEs, we applied a Bayesian model comparison framework[46] to estimate whether the sex-DE effect sizes are consistent with shared, opposite, prenatal-specific, or adult-specific patterns (see STAR Methods; Figure S15). We analyzed the union of the 3,356 sex-DE genes discovered in the two analyses (297 shared genes, i.e., significant sex-DE in both datasets, 2,380 and 679 significant only in the prenatal and adult data, respectively) using this approach. In general, the set of 3,356 sex-DE genes were enriched in brain expression (Wilcoxon test $p = 2.2 \times 10^{-3}$ and $1.9 \times 10^{-19}$ in prenatal and adult brain, respectively) and were less constrained compared with non-sex-DE genes (Wilcoxon test $p = 5.3 \times 10^{-8}$; Figure S16).

In this analysis, two major groups of sex-DEs appeared, covering more than 95% of the detected DE effects. Most of the sex-DE signal, 64.6%, was assigned to prenatal specificity, in line with most of the sex-DE genes originating from the prenatal analysis (i.e., 70.9% of the input genes were sex-DEs only in the prenatal dataset) (Figure 2C). Shared male/female

differences between the two datasets accounted for 30.6% of the effects in total, a considerably greater proportion than estimated based on significant DE genes only (8.3% in the input genes with the same direction of effect). This is, nevertheless, a degree lower than the 64.3% of sex-DE sharing for the same tissue of different individuals (adult cortex, GTEx vs. BrainSeq) (Figure S14C). The great majority of the shared signal (28.1% of all sex-DE genes) was assigned to the model supporting shared effects with a magnitude of effect twice larger in the prenatal brain than in the adult brain (SHARED0.5) (Figure 2C) aligned with the earlier observation of larger sex-DE effect sizes in the prenatal brain.

Only small fractions of the genes were found to have opposite sex-DE effects (of same magnitude) between the two life stages (4.2%) and adult-specific sex-DE (0.7%). The latter finding is striking considering that 20.2% of the input genes were significantly DE only in the adult brain. Given these observations, it appears very little sex-biased expression is captured exclusively in the adult brain. Rather, the early developing brain already displays a considerable fraction of sex-DEs that is maintained into adulthood, albeit with a typically lower degree.

### Shared sex differences in prenatal and adult brain reflect neuronal functions

To allow for the examination of characteristics defining the shared and developmental stage-specific sex-DE, we compiled sets of individual genes that were confidently assigned (gene-specific posterior probability [PP] > 0.8) to one of the sex-DE patterns in the model comparison framework (see STAR Methods; Figure 2D). This approach resulted in the classification of 1,934 genes into the category of prenatal-specific effect (e.g., *DNAI1*, *NPY*, *CTXN3* genes; Figure 2E), 719 into shared effect (a sum of PPs of the three categories of shared effects) (e.g., *IQGAP3*, *POMC*, *SPESP1*, *LINC01597*; Figure 2E), the majority (82.3%, 592 of 719) of which had a larger effect size in the prenatal forebrain (PP.SHARED0.5 > 0.8) and three into adult-specific effect (e.g., *NOX5*; Figure 2E) (Table S6; Figure 2D). In the last category, 45 genes had an opposite direction of effect (e.g., *KCNB1*, encoding part of a potassium channel shown to be highly expressed in neocortical pyramidal cells[47] and linked to encephalopathy, epilepsy, and developmental delay when mutated[48]; Figure 2E). A total of 655 sex-DE genes were left unclassified (PP < 0.8 for any of the models), partly driven by their lower sex-DE effects in the prenatal brain (Figure S17).

We first set out to explain the high degree of shared DE between the prenatal and adult brain, and therefore the lack of adult-specific DE, by investigating their gene expression characteristics and gene functions. As the consistency of the level of expression in the bulk brain tissue appeared not to explain the shared sex-DE pattern (Figure S18A), we investigated whether the shared sex-DE signal might be driven by specific cell types shared but present at different proportions between the prenatal and adult brain (using human brain single-cell RNA-seq data from Song et al.[49]; see STAR Methods). The shared sex-DE genes were more highly expressed in neurons (Wilcoxon test *p* value adjusted for multiple testing = $6.4 \times 10^{-4}$ and $1.9 \times 10^{-3}$ for excitatory and inhibitory neurons, respectively) (Figure 3A) than the prenatal-specific sex-DE genes, offering a plausible

explanation of their consistency across the two studied life stages. GO analyses further indicated neuronal functions contributing to the shared sex-DE between the prenatal and adult brain (Table S3), as the shared sex-DE genes with male-biased expression displayed enrichments in terms linked to neuron and synapse activity (45 of 146 significant GO terms).

The prenatal-specific sex-DE genes appeared shared across different cell types, as they were more highly expressed in progenitor cells (Wilcoxon test *p* value adjusted = $4.9 \times 10^{-7}$ and $1.4 \times 10^{-3}$ in neuronal and oligodendrocyte progenitor cells, respectively) (Figure 3A) and other non-neuronal cells (Figure S18B) compared with the shared sex-DE genes. In agreement with these findings, the prenatal-specific sex-DE genes were more broadly expressed across adult human tissues from GTEx (Wilcoxon test *p* = $6.7 \times 10^{-27}$) and less tissue-specific (Wilcoxon test *p* = $9.0 \times 10^{-36}$) than the shared genes (Figures S18C and S18D), suggesting a larger diversity of functions of these genes. However, no differences in gene constraint were seen between the two groups (Figure S18E). In addition, a cell-type-specific sex-DE analysis with TOAST[50] (see STAR Methods) showed that a large fraction (*n* = 628, 19.71%) of the prenatal sex-DEs could be driven by effects originating from neuronal progenitor cells (NPCs), the cell type with the most sex-DE effects (*n* = 2,697 in NPC, *n* = 330 in excitatory neurons).

### Early prenatal testosterone surge as potential driver of prenatal-specific sex differences in expression

We next set out to understand the biological processes relating to the sex-DE genes specific to the prenatal life stage. Since the studied time span coincides with the first testosterone surge (proposed to occur between 9 and 18 PCW[51,52]), we hypothesized that hormonal factors could contribute to sex-DEs in early development. To test for this, we analyzed the transcription factor binding sites (TFBSs) enriched in these genes' promoters using the UniBind framework[53] (see STAR Methods).

While no TFBSs were significantly enriched (q value < 0.05) in the genes that showed shared sex-DE throughout lifespan, the prenatal-specific genes displayed several enrichments, with female-biased genes enriched in eight TFBSs, and the male-biased ones in 74 TFBSs. Notably, these enrichments included established hormonal transcription regulators like androgen receptor (*AR*) (q value = $2.7 \times 10^{-4}$, enrichment odds ratio in male-biased genes = 1.5) and Estrogen Receptor 1 (*ESR1*) (q value = 0.03, enrichment odds ratio in male-biased genes = 1.3) (Table S7). Moreover, some of the 74 TFBSs enriched for male-biased genes were known *AR* interactors, like *FOXA1* and *FOXP1*,[54] or *ESR1* interactors, like *JUN* (from the Reactome database[55]). Interestingly, *AR* expression was female-biased in the prenatal brain (prenatal logFC = 0.32; adult logFC = 0.05). This upregulation of *AR* in female compared with male individuals is in line with the known auto-downregulation of *AR* in presence of androgens.[56]

The enrichment of prenatal-specific sex-DE genes in androgen-responsive genes was further validated using a set of genes shown to be differentially regulated with androgen treatment[57] (Figure S19). We observed a more than 2-fold enrichment of female-biased genes in genes upregulated with testosterone treatment in NSCs (relative enrichment = 2.61,

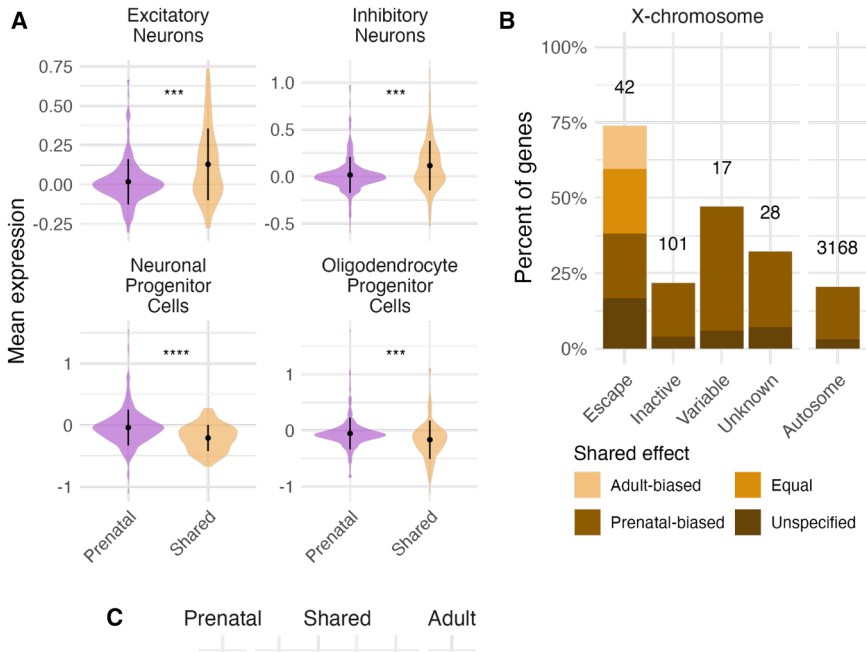

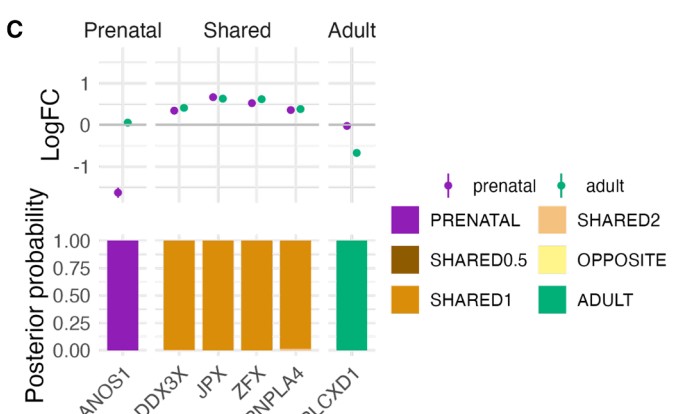

**Figure 3. Characteristics of the shared and prenatal-specific sex-DE genes**

(A) Expression of the shared and prenatal-specific genes in excitatory neurons, inhibitory neurons, neuronal progenitor cells, and oligodendrocyte cells. Difference in mean expression between the two categories was tested with a Wilcoxon test. *** and **** indicate Wilcoxon test $p$ value $\leq$ 0.001 and $\leq$ 0.0001, respectively.

(B) Percentages of genes in each shared category (shared with the same or different effect size) for different classes of X chromosome inactivation (XCI) genes (escape, inactive, variable, or unknown) and autosomal genes. The number of genes per category is shown as the top of each bar.

(C) Examples of XCI escape genes classified as prenatal-specific ($n = 1$), shared ($n = 4$), or adult-specific ($n = 1$). The upper plot represents the logFC (with standard error) in both datasets. The lower plot shows the posterior probabilities from the Bayesian model for each gene.

hypergeometric test $p = 4.20 \times 10^{-8}$), while the prenatal-specific male-biased genes were enriched in genes downregulated by the same testosterone treatment (relative enrichment = 1.85, hypergeometric test $p$ value = $9.75 \times 10^{-23}$). Similar enrichment patterns were observed for other androgen treatments (e.g., DHT). In contrast, the shared sex-DE genes were not significantly enriched in genes responding to androgen treatments. This points to the specificity of the hormonal effects to the prenatal stage, potentially related to the early testosterone surge. The downregulation, rather than upregulation, of male-biased genes and the upregulation of female-biased genes likely reflect the complex regulatory mechanisms of steroid hormones in brain sexual differentiation, including AR autoregulation and the aromatization of testosterone to estradiol.

## Lifelong gene expression sex bias due to escape from X chromosome inactivation

Given the early establishment of X chromosome inactivation (XCI)[58] and the reported adult tissue similarities in the sex-biased expression introduced by escape from XCI,[59] we reasoned that the X chromosome genes that escape from XCI would display

high sharing of sex-biased expression also across life stages. Indeed, known XCI escapees ($n = 42$ with sex-DE, 22.2% of X chromosome sex-DE genes) were more concordant in sex-DEs in the prenatal and adult brain in comparison with autosomal ($n = 3168$) and inactive X-linked genes ($n = 101$). Altogether, 73.8% (31 of 42) of the escapees were classified as shared sex-DE genes compared with 21.8% (22 of 101) of known inactive X chromosome genes ($\chi^2$ $p$ value = $1.37 \times 10^{-8}$ for difference in proportions) (Figure 3B). Further, XCI escapees were typically highly consistent in their degree of sex bias between the two life stages, with 21.4% (9 of 42) of the escape genes (vs. 0 of 101 inactive genes, $p$ value = $9.52 \times 10^{-6}$) displaying similar effect sizes in the prenatal and adult brain (PP.Shared1 >0.8), including *DDX3X* (PP.Shared1 = 0.997), *JPX* (PP.Shared1 = 1.000), and *PNPLA4* (PP.Shared1 = 0.986), all female-biased (Figure 3C).

Interestingly, *ZFX*, a known XCI escapee and a transcription factor proposed to influence autosomal sex-DE,[60] was consistently female-biased across the developing and adult brain (prenatal logFC = 0.52, adult logFC = 0.62, PP.Shared1 = 1). Supporting a role in autosomal gene regulation, *ZFX* binding sites were enriched in the prenatal male-biased genes (q value = 0.02, enrichment odds ratio = 1.3; Table S7). The enrichment of *ZFX*-regulated genes among the male-biased genes, rather than the female-biased ones, could be explained by *ZFY*, a homologous gene in the male-specific region of the Y chromosome (highly male-biased, prenatal logFC = −8.69 and adult logFC = −8.43), known to bind to similar motifs and occupy similar genomic locations than *ZFX*.[60]

A few escapees ($N = 11$ with PP.Shared <0.8) displayed more divergent patterns of sex-DEs. *PLCXD1*, a gene in the

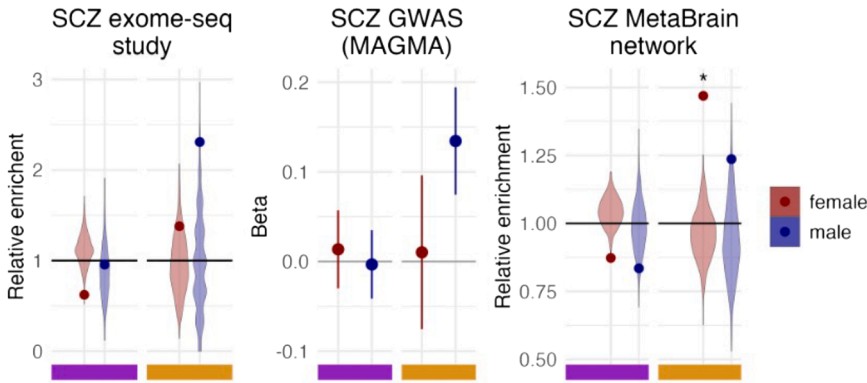

**Figure 4. Enrichment analysis of three different schizophrenia gene lists in the shared and prenatal-specific sex-DE genes**

(Left) Genes from an exome-sequencing study of schizophrenia (SCZ) (n = 244). (Middle) Genes from an SCZ GWAS analyzed with MAGMA. (Right) Genes from an SCZ co-regulation network analysis from the MetaBrain resource (n = 2,737). The colored box at the bottom indicates which data the results are from (violet = prenatal and yellow = shared). The violin plots on the left and right plots represent the distribution of the relative enrichment for the 1,000 random gene lists and the point the enrichment for the true gene list. The error bars in the middle plot depict the SE of the beta values. *Significant enrichment of the SCZ gene list based on both multiple testing corrected p values from hypergeometric test for the gene list of interest and a permutation test against random gene sets matched for the level of gene expression as defined in STAR Methods.

pseudo-autosomal region of the X chromosome, was the only escape gene with evidence for adult-specificity of sex-DEs with large male-biased effects (logFC = −0.68, PP.Adult = 1.00) (Figure 3C). Four escape genes displayed evidence for prenatal specificity. These include *ANOS1* that was highly male-biased in expression, unusually for an escape gene in the non-pseudo-autosomal region,[59] in the prenatal brain (logFC = −1.63, PP.Prenatal = 1.00) (Figure 3C). *ANOS1* is crucial for neuronal migration in the developing brain[61] and was previously suggested as a gene subject to variable escape in adult human tissues.[59] These variable patterns of escape can point to cell-type-dependent control of XCI but can also reflect, e.g., hormonal or other age-related regulation of gene expression dampening the sex bias arising from escape. Further, 25 previously established escape genes did not show significant sex bias in either of the life stages, possibly due to their lower level of expression in the brain (Figure S20A), these genes also being less frequently observed as sex-biased in adult human tissues than other escape genes (Figure S20B).

### Genes with shared sex bias associate with disease networks

To understand the potential phenotypic relevance of these life-long and life-stage-specific sex biases in the forebrain gene expression, we examined the enrichment in overlap of the observed sex-DE patterns with genes implicated in the etiology of brain disorders identified through both exome-sequencing and genome wide association studies (GWAS).

Across the 19 unique diseases studied we found little conclusive evidence for enrichment in overlap of sex-DE and brain disease-associated genes in any of the sex-DE gene lists (Figure S21; Table S8). For the nine gene lists derived from exome-sequencing studies, these generally representing high-impact disease genes where the functional loss of another gene copy increases in the disease risk, we observed no significant enrichment (hypergeometric test p value < 6.9 × 10⁻⁴ = 0.05/9 × 8) with the prenatal-specific or shared sex-DE genes. Four disease gene lists displayed a nominally significant enrichment for sex-DEs and these findings were further validated by a comparison against a background of genes with a similarly high

expression level in the brain (see STAR Methods). For instance, genes implicated in schizophrenia (244 genes) showed an enrichment among the male-biased shared sex-DE genes (2.3 relative enrichment, hypergeometric p value = 0.029, permuted p value = 0.004; Figure 4). The overlap with GWAS associations, where the association is typically mediated through regulatory effects, similarly found no significant enrichments (MAGMA p value < 6.9 × 10⁻⁴ = 0.05/9 × 8) in small GWAS p values at and near the sex-DE genes. Some of the nominally significant findings, nevertheless, corroborated the earlier results, e.g., male-biased shared sex-DE genes were enriched in schizophrenia associations (MAGMA p value = 0.012; Figure 4). Such limited direct disease overlap can be at least partly expected, as genes with phenotypic effects tend to tolerate less genetic and expression variation,[62–64] in contrast to sex-DE genes that, in general, were less constrained than genes without sex differences (Figures S8B and S16A). A similar lack of significant enrichment was observed when focusing on the most constrained genes (i.e., three lowest constraint bins; Figure S16B).

To confirm the limited overlap with disease-associated genes, we performed an enrichment analysis on gene lists for 16 neuropsychiatric disorders compiled by Mulvey and colleagues.[65] Here, in agreement with our previous observations, we observed a lack of enrichments for these genes lists (Figure S22). This is, however, in contrast with the findings from the original study,[65] but the difference in findings can be attributable to their analysis focusing on very specific sub-regions of the forebrain (ventromedial hypothalamus and arcuate) with the help of spatial transcriptomic technology. To validate that our sex-DE findings are, nevertheless, consistent with prior literature, we replicated the previously reported enrichments of sex-DE genes with those differentially regulated in an ASD case-control study,[23] e.g., we observed that shared female-biased genes were enriched in genes downregulated in ASD cases (Figure S23).

As sex-biased expression appeared not to directly impact disease genes, we hypothesized that the sex-DE genes could rather exert phenotypic effects through being interaction partners to disease-associated genes. To test for these more distal effects, we used the brain-specific gene co-regulation networks from the MetaBrain resource.[66] Application of this network has

pinpointed sets of genes highly likely to be co-regulated with genes within the schizophrenia, multiple sclerosis (MS), Parkinson's disease, Alzheimer's disease, or amyotrophic lateral sclerosis GWAS loci (see STAR Methods). We observed that the shared female-biased sex-DE genes were significantly enriched (hypergeometric test $p$ value $< 7.8 \times 10^{-4} = 0.05/8 \times 4$ and permuted $p$ value $< 0.05$) in the cortex-specific co-regulatory networks of both schizophrenia (Figure 4) and MS GWAS genes (Figure S24A; Table S8) ($p$ value $= 3.21 \times 10^{-17}$ and $1.36 \times 10^{-6}$, respectively). Additionally, the adult sex-DE genes were significantly enriched in the co-regulatory network of MS GWAS genes (Figure S24B). The association with schizophrenia was not driven by the GWAS genes themselves, as no enrichment was detected in overlap with schizophrenia GWAS genes from MetaBrain (e.g., for shared male-biased genes, $p$ value $= 0.06$). While the direction of the sex bias effect for a co-regulation network is difficult to translate to the direction of the impact on the phenotype, these findings, nevertheless, suggest phenotypic relevance of the brain gene expression sex biases may arise through the modulation of the activity of the gene networks involved in mediating the genetic disease risk.

## DISCUSSION

To better understand the mechanisms underlying the widespread sex biases observed in brain structure, function, and various brain-related conditions, we set out to characterize sex differences in the human brain transcriptome during early development and adulthood. Using data from the HDBR, that spans the period of neurogenesis in prenatal development (5–17 PCWs), and from adult brain samples from the GTEx project, we discovered that transcriptional sex differences are widespread and systematically present already in the early brain development and that a large fraction of these sex biases are maintained into adulthood in a manner where most of the adult brain sex differences are already present in the prenatal brain. Overall, our work demonstrates the relevance of fetal development to sex differences and highlights the roles of sex-chromosomal and early hormonal influences in shaping sex-biased brain biology.

While sex biases in the adult brain have been extensively studied, both in terms of gene expression and its genetic regulation,[15,67] the scarcity of early brain samples and the heterogeneity of the available datasets has posed limitations to the study of sex differences in the fetal brain. To facilitate the discovery of sex-DEs in the prenatal brain samples from the unique HDBR resource,[33] we accounted for the high transcriptional variability by focusing our assessments on the forebrain region and by modeling the active developmental period using pseudotime, which allowed us to better align the samples across the developmental time points than the ordinal categories typically used. With this approach, our analyses revealed an abundance of sex-DE genes during the early development, providing support for the early origins of brain sex differences.

Although abundant, the detected sex differences in gene expression were mostly small in scale, comparable to the allelic effects of common gene regulatory variants,[68] aligned with findings from adult tissue.[15] This finding implicates that the devel-

oping brain tolerates a limited amount of variability between groups of normally developing individuals. Rather, as sex-DE typically impacted genes more tolerant to genetic variation, many crucial brain processes can in fact be shielded from expression variation between the sexes.

Comparison of prenatal sex-DEs to that from the adult forebrain uncovered two dominant patterns of sex-DEs, prenatal-specific, and life-stage-shared sex effects. The apparent lack of adult-specific sex-DE suggests the sex biases detected in the adult brain are largely emerging earlier in brain development further pointing to the relevance of early development. This finding contrasts with recent evidence from Rodríguez-Montes and colleagues,[69] where sexual maturity was found as the crucial time point for the onset of sex-biased expression. We attribute these differences to the larger sample size in the current study with respect to the fetal time point, and our approach that considers the similarity of sex effects beyond those genes that display significant sex differences at a given life stage. Mechanistically, the consistency of sex-DEs may be driven by the shared neuronal cell types between the prenatal and adult brain.

As the effect of the environment is likely small in the very early stages of brain development, we hypothesize that the sex biases in the fetal human brain primarily originate from intrinsic biological factors, such as direct sex chromosome effects or early hormonal influences.[70] Indeed, we found support for both processes shaping transcriptional sex differences, partly in a life-stage-dependent manner. Following the finding that sex-DE genes specific to the fetal stages are enriched for transcription factor binding sites of hormone receptors, we propose some of the observed sex effects are reflections of the significant hormonal changes occurring during this developmental window owing to the fetal testosterone surge. This suggests the transient influence of the early testosterone surge extends beyond sex determination and genitalia formation, potentially leading to stable sex differences in the organization and function of the brain.

Sex-chromosomal contributions to sex-DEs were evident in the clear sex biases of genes escaping from XCI, in line with earlier reports.[15] The high consistency of sex effects at XCI escapees between the prenatal and adult brain confirms the early establishment of XCI and escape,[71,72] adds to the known similarities in escapee expression sex biases in adult tissues,[59] and points to escape from XCI being a highly stably maintained phenomenon throughout life irrespective of cell type compositional, hormonal, and other changes. As XCI escapees have proposed roles in the regulation of autosomal gene expression,[60,73] the substantial and consistent sex-DEs associated with escape raises the possibility of escapees introducing broader sex effects in the brain beyond the sex-chromosomal expression.

Understanding how the life-stage-specific and shared sex effects in gene expression relate to phenotypic sex differences is a key open question. Despite the similar processes implicated by sex-differential expression and genetic studies of neurodevelopmental disorders, we detected limited overlap between sex-DE and disease genes, echoing recent observations with regard to fetal cortex sex-DE and ASD risk genes.[26] Focus on genes linked to diseases through genetic studies allowed us to examine

genes with confident links to the disease etiology rather than genes identified through case-control comparisons of gene expression (e.g., ASD-related alterations[74]) that could rather reflect consequences of the disease process. We attribute the lack of overlap sex-DE and disease genes to systematic differences between sex-DE and GWAS signals, e.g., in terms of gene constraint, in analogy to recent findings on the distinct characteristics of common gene regulatory variation and GWAS hits.[64] In the light of our findings, we propose that the brain sex-DEs may nevertheless play a role in differential disease susceptibility through impacting the networks that the disease genes are involved in.

## Limitations of the study

We compared patterns of sex-biased gene expression from two life stages: prenatal development and adulthood. Other time points across development would allow building a clearer picture of the life course dynamics of sex-DEs, for instance, in relation to the onset of sex-DEs and the impact of other critical windows for gonadal hormone differences. Projects like the Developmental GTEx[75] will be key for closing these gaps in knowledge. The smaller sex-DE effect sizes in the adult brain are a potential reflection of the increased expression variability associated with aging[76] induced by environmental influences we are unable to fully account for in the analysis. This may complicate the inference of the proportions of genes that show sex-DEs across the lifespan, or only in adulthood, and lead to an underestimation of these proportions. Also, we currently lack an understanding of the potential consequences of the prenatal-specific sex-DEs. The study of the co-regulatory network genes might bias our findings toward shared genes as the networks are constructed from adult tissue. Finally, we used pseudotime inference to provide us with a continuous ordering of the HDBR samples along development to boost power for sex-DE discovery. Pseudotime in HDBR and particularly in GTEx does not, however, solely reflect sample development or age, but to a degree also captures cell-type differences and technical variability.

## Conclusions

We present a comprehensive evaluation of the emergence and dynamics of transcriptomic sex differences in the developing and adult forebrain. Our findings highlight the fetal development as a crucial time point for the introduction of brain-related sex differences and propose the involvement of both hormonal and sex-chromosomal contributions in shaping the brain sex biases. Although it remains unclear how the identified transcriptional sex biases are mechanistically linked with brain phenotypes, our findings point to the direction that a degree of the sex differences in brain-related traits stem from non-environmental effects originating from the prenatal period. This can have implications for the understanding of sex-biased disease susceptibility and the development of more targeted diagnostics and therapeutics.

## RESOURCE AVAILABILITY

### Lead contact

Requests for further information and resources should be directed to and will be fulfilled by the lead contact, Taru Tukiainen (taru.tukiainen@helsinki.fi).

### Materials availability

This study did not generate new unique reagents.

### Data and code availability

All raw data used in this article were previously generated by HDBR and GTEx and are publicly available in the EBI ArrayExpress database under accession number E-MTAB-4840 and in the GTEx online portal https://www.gtexportal.org/home/, respectively. The processed data supporting the conclusions of this article are included in the supplementary tables. All original code used in this study is available at GitHub https://github.com/cbenoitp/sexDE_prenatal_brain and Zenodo https://doi.org/10.5281/zenodo.15039762.

## ACKNOWLEDGMENTS

T.T. was funded by the Sigrid Jusélius Foundation (https://sigridjuselius.fi/en/), the HiLIFE Fellows Program, and the Research Council of Finland (https://www.aka.fi/en/) grant numbers 315589 and 320129. Open access was funded by Helsinki University Library. The graphical abstract was created with BioRender.

## AUTHOR CONTRIBUTIONS

The project was conceived and planned by C.B.-P. and T.T. C.B.-P. performed the analyses under the supervision of T.T. J.V.A. conducted the MAGMA analysis and J.K. preprocessed the prenatal dataset. C.B.-P. and T.T. wrote the manuscript, with J.T.L., J.V.A., and M.J.D. providing comments and edits.

## DECLARATION OF INTERESTS

M.J.D. is a founder of Maze Therapeutics.

## STAR★METHODS

Detailed methods are provided in the online version of this paper and include the following:

- KEY RESOURCES TABLE
- EXPERIMENTAL MODEL AND STUDY PARTICIPANT DETAILS
  - Data
  - HDBR dataset
  - GTEx dataset
  - BrainSeq dataset
- METHOD DETAILS
  - Principal component analysis
  - Pseudotime analysis
  - Cell type decomposition analysis
  - Normalization of expression counts
- QUANTIFICATION AND STATISTICAL ANALYSIS
  - Sex differential expression analysis
  - Prenatal cell-type specific sex-DE analysis
  - Comparison of sex-DE genes across life stages or datasets
  - Bayesian method to compare sex-DE across dataset
  - Analysis of the sex-DE genes properties
  - Enrichment analyses
  - GO term over-representation analysis
  - Enrichment for transcription factor binding sites in promoters
  - Enrichment for androgen-responsive genes
  - MAGMA analysis
  - Hypergeometric enrichment analysis
  - Enrichment analysis of Mulvey et al. 2024[65] gene lists
  - Enrichment analysis of ASD case-control gene list
  - Replication analysis of prenatal sex-DE genes

## SUPPLEMENTAL INFORMATION

**CellPress**

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

CellPress

## STAR★METHODS

### KEY RESOURCES TABLE

| REAGENT or RESOURCE | SOURCE | IDENTIFIER |
|---|---|---|
| **Deposited data** | | |
| Analyzed prenatal data (HDBR) | Lindsay et al.[33] | EBI ArrayExpress database: E-MTAB-4840 |
| Analyzed adult data (GTEx v8) | GTEx consortium | https://www.gtexportal.org/home/ |
| **Software and algorithms** | | |
| All code to reproduce analyses presented in this work | This paper | https://github.com/cbenoitp/sexDE_prenatal_brain; https://doi.org/10.5281/zenodo.15039762. |
| Salmon | Patro et al.[77] | https://github.com/COMBINE-lab/salmon |
| CIBERSORTx | Newman et al.[37] | https://cibersortx.stanford.edu/ |
| UniBind | Puig et al.[53] | https://unibind.uio.no/ |
| MAGMA | De Leeuw et al.[78] | https://cncr.nl/research/magma/ |
| phenopath | Campbell et al.[39] | https://doi.org/10.18129/B9.bioc.phenopath |
| edgeR | Robinson et al.[79] | https://doi.org/10.18129/B9.bioc.edgeR |
| variancePartition | Hoffman et al.[40,41] | https://doi.org/10.18129/B9.bioc.variancePartition |
| sva | Leek et al.[80] | https://doi.org/10.18129/B9.bioc.sva |
| Qvalue | Storey et al.[81] | https://doi.org/10.18129/B9.bioc.qvalue |
| linemodels | Pirinen[46] | https://github.com/mjpirinen/linemodels |
| clusterProfiler | Yu et al.[82] | https://doi.org/10.18129/B9.bioc.clusterProfiler |

### EXPERIMENTAL MODEL AND STUDY PARTICIPANT DETAILS

#### Data

Three different public RNA-seq datasets were used in this study: Human Developmental Biology Resource (HDBR),[33] Genotype-Tissue Expression project (GTEx)[35] and BrainSeq.[83] The number of samples per dataset, per tissue and age group (pre- or postnatal) are detailed in Table S1.

#### HDBR dataset

The HDBR dataset contains 542 prenatal RNA-seq samples from 187 distinct individuals. These individuals were previously confirmed to be karyotypically normal,[33,34] and from previous publication using SNP array data from HDBR,[84] we do not expect close relatives among these individuals. For our analysis, we selected only the samples coming from the forebrain (specific region of the forebrain, forebrain fragment, or whole forebrain). We defined the sex of the individuals by checking the number of reads mapped to genes known to be expressed in one sex only: the protein-coding genes of the Y chromosome (only expressed in males) and *XIST* gene (only expressed in females). A cutoff of 7000 was used to separate male and female samples (Figure S25A). Meaning that a sample was labeled male if the count on chromosome Y protein-coding genes was greater than 7000 and the count on *XIST* gene lower than 7000. And vice versa, a sample was labeled female if the count on chromosome Y protein-coding genes was lower than 7000 and the count on *XIST* gene greater than 7000. All samples for which we could not assign sex using these criteria were removed from our analysis. We also removed the only sample from 4PCW and one outlier sample for which the counts were different from all other samples (from Pearson's correlation analysis). The final number of samples used is 266 corresponding to 72 distinct individuals (Figure S1A). The developmental stages we will be referring to in this work correspond to the number of weeks post-conception (PCW) assigned by the HDBR consortium to each fetus. The age of the samples spans from 5 to 17 PCW covering the second half of the first trimester of pregnancy and the beginning of the second trimester. The developmental stages of the earliest samples (up to 60 days post conception) were annotated by HDBR using Carnegie stages, and the different Carnegie stages were merged into PCW. We then divided the samples into 8 different developmental stages as follows: 5-7PCW, 8PCW, 9PCW, 10PCW, 11PCW, 12PCW, 13PCW, and 14-17PCW.

#### GTEx dataset

GTEx contains postmortem adult samples from a wide range of tissues. The forebrain RNA-seq samples were used to assess sex-DE in adult. 1633 samples (434 female and 1199 male samples) from 337 distinct individuals aged between 20 and 79 years and spanning

8 different forebrain regions (amygdala, anterior cingulate cortex, caudate, frontal cortex, hippocampus, hypothalamus, nucleus accumbens, and putamen) were used (Figure S1B). Again, the sex of the samples was confirmed using the counts data of *XIST* and the protein-coding genes on the Y chromosome (Figure S25B). In this data, a threshold of 150 for the count of *XIST* and 1000 for the sum of the counts of the protein-coding genes on the Y chromosome.

### BrainSeq dataset
The data from the BrainSeq (phase 1) dataset was used to assess the replicability of the sex-DE genes in the adult cortex. For this replication analysis, we selected only the 189 samples (134 male and 55 female) from dorsolateral prefrontal cortex (DLPFC) of control adults.

## METHOD DETAILS

### Principal component analysis
The principal component analysis (PCA) was carried out using ade4 R package[85] on a total of 4137 samples from pre- and postnatal origin, as well as different tissue origins (Table S1). The samples came from the 3 datasets used in the main text (HDBR, GTEx, and BrainSeq) and the EvoDevo mammalian organs dataset.[86] This last dataset comprises 254 samples from 94 individuals spanning different tissues (brain, liver, heart, and kidney) and different life stages (from prenatal to adulthood). This dataset was added in the PCA analysis to ensure we were not seeing an effect of the life stage confounded by the dataset of origin. Indeed, EvoDevo is the only dataset containing both prenatal and postnatal samples for each tissue.

The raw counts were first normalized using the variance stabilizing transformation (vst) function from DESeq2.[87] Then, we removed the effect of the study of origin using the batch correction method removeBatchEffect from limma.[88]

### Pseudotime analysis
For each sample, the pseudotime was inferred using the 2000 most highly variable genes with the phenopath R package.[39] The expression values were formatted as $\log_2(TPM+1)$ and the following parameters were used: elbo_tol = $1 \times 10^{-6}$, thin = 20 and design matrix = ∼Sex. This analysis uses sex as a covariate allowing us to detect different trends in pseudotime between males and females. We checked that the function had reached convergence using the plot_elbo function. The set of genes used to infer pseudotime was removed from all analyses related to pseudotime (i.e., pseudotime-DE analysis).

To compute the Kendall correlation between the inferred pseudotime and the developmental stages, we converted the categorical variable corresponding to the developmental stages to a continuous variable.

We conducted several sensitivity analyses to test the robustness of the pseudotime inference: 1. We tested in HDBR how changing the number of input genes (from 100 to 3000 most variable genes) in the pseudotime analysis changes the inferred pseudotime (Figure S5). 2. We inferred pseudotime with another method, Monocle2 (Figure S4). We used Monocle2 following the "Constructing single cell trajectory" tutorial from this page: http://cole-trapnell-lab.github.io/monocle-release/docs/#trajectory-step-1-choose-genes-that-define-a-cell-s-progress. First, we selected the top 1000 genes DE with the developmental stages to use to construct the trajectory. Next, we reduced the dimensionality of the data controlling for the forebrain region as we did not want it to contribute to the trajectory. 3. We inferred pseudotime with including the reported developmental stages as a covariate in the phenopath algorithm. 4. We inferred pseudotime using only one sample per individual. The selected sample was in priority from the telencephalon (region with the highest number of samples), if any was available. 5. We tested the impact of the sex covariate on the inferred pseudotime by permuting the sex labels across individuals and conducting the pseudotime analysis and the subsequent pseudotime-DE analysis, repeating this 50 times. We additionally correlated the inferred pseudotimes in HDBR and GTEx with available technical covariates.

### Cell type decomposition analysis
We used CIBERSORTx[37] to do a cell type decomposition of the prenatal and postnatal forebrain samples. This tool uses single-cell RNA-seq (scRNA-seq) data to infer the transcriptomic signature of each cell type and then estimates the abundance of each cell type in the bulk RNA-seq samples. We used 3 publicly available scRNA-seq datasets present in the STAB database[49] as the reference. The 3 chosen datasets[89–91] contained samples from embryonic (5–26 PCW) and adult (22–63 years) individuals and spanned different regions of the forebrain (Table S1). We used the docker container to run CIBERSORTx with the following parameters: –rmbatchSmode TRUE –perm 100 –replicates 5 –sampling 0.5 –fraction 1 –k.max 999 –q.value 0.01.

### Normalization of expression counts
For the HDBR forebrain dataset, transcripts abundance was assessed with Salmon[77] using Gencode (version 28) reference annotation. Counts for each transcript of a gene were summed to obtain the gene quantification. These counts were then normalized using EdgeR[79] TMM normalization. Only genes with count per million (CPM) values higher than 1 in more than 10 samples were kept for the differential analysis. Finally, the normalization was done with voomWithDreamWeights function from variancePartition R package[40,41] using sex, pseudotime, individuals, and surrogate variables (svas) as covariates. The svas were computed with sva R package.[80]

For the GTEx dataset, we used the raw counts (using version 26 of Gencode annotation) publicly available on the online portal (https://www.gtexportal.org/home/). We did the same TMM normalization with EdgeR, applied the same filter on CPM values, and carried out the normalization with voomWithDreamWeights using sex, pseudotime, cause of death, individuals, and svas as covariates.

## QUANTIFICATION AND STATISTICAL ANALYSIS

### Sex differential expression analysis

We tested for genes differential expression between males and females defined as a binary traits using the linear mixed model for repeated measures (dream) from the variancePartition package,[40,41] adjusted for covariates, i.e., pseudotime, pseudotime/sex interaction, and surrogate variables as fixed effects, and accounting for multiple samples from the same individuals as a random effect.

In the forebrain HDBR analysis, the following model was used:

$$expression \sim sex + pseudotime + sex : pseudotime + (individual) + svas$$

In the forebrain GTEx analysis, the following model was used:

$$expression \sim sex + pseudotime + sex : pseudotime + forebrain\ region + cause\ of\ death + (1|individual) + svas$$

For both datasets, we extracted three lists of DE genes.

(1) Genes differentially expressed between males and females (sex-DE)
(2) Genes differentially expressed along pseudotime (pseudotime-DE)
(3) Genes with a significant interaction between sex and pseudotime (interaction-DE)

For all genes in each analysis, the false discovery rate (q value) was computed using the qvalue R package[81] to correct for multiple testing. In all these analyses, a gene was considered significantly differentially expressed if it passed the following thresholds: $q$ value $<0.01$.

Sex differential expression refers to group-level average differences between individuals defined as males or females in our analysis.

### Prenatal cell-type specific sex-DE analysis

We carried out a differential expression analysis on the prenatal data at the cell-type level using the TOAST method.[50] We used a model including sex and pseudotime as covariates in addition to the cell type. The individual covariate could not be included as TOAST does not allow for random effect variables in the model. As recommended by the authors of this method, we also did a permutation analysis (1000 permutations on the sex phenotype) to identify "real" effects. We filtered the significant sex-DE genes per cell type (FDR$<0.05$ and permutation $p$ value$<0.05$).

### Comparison of sex-DE genes across life stages or datasets

To compare the results of the differential analysis between the two life stages studied, we first look at the overlap of the sex-DE genes ($q$ value $<0.01$). A hypergeometric test was used to assess the significance of the overlap. For the comparison not to be too impacted by the choice of the thresholds and by the power of each dataset, we also computed the replication rate ($\pi_1$) using the qvalue R package[81] with lambda = 0.5.

We also computed a consistency metric for the sex-DE. This metric represents the percentage of genes that vary in the same direction in the sex-DE analysis (the sign of the t-statistic is the same in both analyses). We tested if the consistency was different than what was expected by chance using a binomial test. The consistency of the sex-DE effect size was also assessed with Pearson's correlation and visualized in a scatterplot.

These analyses were performed on all sex-DE genes and then, we separated the genes into three groups: autosomal genes, genes escaping XCI, and inactive X chromosome genes as defined in.[59]

Similar metrics were computed to compare two datasets of the same tissue and same life-stage: GTEx frontal cortex (BA9) and BrainSeq DLPFC.

### Bayesian method to compare sex-DE across dataset

To compare sharing and specificity of sex-DE genes between prenatal and adult brain, we used the *linemodel* Bayesian method.[46] For all sex-DE genes (prenatal or adult), we estimated probabilities between six possible explanations of the observed data: the effect is present in only one of the dataset but not in both (PRENATAL *slope* = 0 and ADULT *slope* = Inf); the effect is shared by the datasets with the same effect size (SHARED1 *slope* = 1); the effect is shared with a higher effect size in prenatal brain (SHARED0.5 *slope* = 0.5); the effect is shared with a higher effect size in adult brain (SHARED2 *slope* = 2); or the datasets have opposite effect sizes (OPPOSITE *slope* = −1) (Figure S15). The prior variance of effect size (scale) was set to 0.1971916 for all models which correspond to the 95th percentile of the logFC in the prenatal dataset divided by 2. We set the correlation parameter (*cor*), which corresponds to the allowed

deviation from the expectation to 0.995 for all models except for OPPOSITE for which *cor* = 0.990. Lastly, the correlation (*r.lkhood*) between the estimators of the effect sizes between the two dataset was set to 0 as the datasets are independent (i.e., they don't contain any overlapping individuals). We used the *line.models.with.proportion* function that gives for each gene a posterior probability on each model as well as the overall proportions of effects coming from each model with 2000 iterations and a burn-in period of 200 iterations.

For each gene, we then classified them into 4 categories using threshold on the posterior probability: shared effect (SHARED0.5 + SHARED1 + SHARED2 > 0.8), prenatal-specific effect (PRENATAL >0.8), adult-specific effect (ADULT probability >0.8) and opposite effect (OPPOSITE >0.8).

This Bayesian approach with the same parameters was also used to compare sex-DE results of all GTEx tissues against each other as well as the GTEx forebrain dataset against the BrainSeq dataset. For the comparison of GTEx tissues, to account for the fact that some samples from different tissues could come from the same individual, we computed a correlation between the pairs of tissues. For each pair of tissue, the *r.lkhood* parameter was computed by taking the Pearson correlation between the logFC of all genes that are not strongly sex-DE (*p* value >0.10) in both tissues.

### Analysis of the sex-DE genes properties
We looked at several properties of the sex-DE genes: their level of expression in both bulk and single-cell data, their level of constraint, their breadth of expression, tissue specificity and consistency in sex-DE across adult human tissue (GTEx).

For the comparison of genes level of expression, we used expression from bulk RNA-seq data in adult and prenatal brain samples from BrainSpan (www.brainspan.org). We also used the genes expression from the scRNA-seq datasets from the STAB database[49] previously describe, allowing to look at expression in different cell types. For the comparison of genes constraint, we used the LOEUF score describe in[62] downloaded from Gnomad (v4.1) browser. In addition, we used several metrics derived from the full GTEx (v8) dataset containing gene expression for 52 human adult tissues. We compared the breadth of expression, i.e., in how many tissues a gene is expressed in, the tissue specificity (tau value) computed by Palmer and colleagues,[92] and the consistency of sex-DE across tissues, i.e., in how many tissues a gene is sex-DE with the same direction of effect (male- or female-biased).

### Enrichment analyses
For all enrichment analyses requiring a list of significant genes, we tested each DE analyses separating the significant DE genes (*q* value <0.01) in two lists corresponding to the direction of effect (either positive or negative logFC). When analyzing the sex-DE genes, the genes used to assign sex to the samples were removed. For the analyses on the Bayesian results, we used the classification of the genes created using the posterior probabilities.

### GO term over-representation analysis
The over-representation analysis was performed with compareCluster function from the clusterProfiler R package for each GO category separately (Biological process, Molecular Function and Cellular Component).[82]

### Enrichment for transcription factor binding sites in promoters
We used the UniBind enrichment tool[53] to predict if promoters of DE genes were enriched in transcription factor binding sites (TFBS). Promoters were defined as the region spanning 2kb upstream of the gene transcription start site similarly to.[15] We used the subcommand oneSetBg of the UniBind_enrich script with the promoters of all genes analyzed in the differential analysis as background and the robust human TFBS sets computed by Unibind as a database.

### Enrichment for androgen-responsive genes
We performed an enrichment analysis of the androgen-responsive genes from Quartier and colleagues[57] in the prenatal-specific and shared sex-DE genes defined with the Bayesian model. For these analyses, we separated the genes lists by direction of effect (female- or male-biased). We tested the enrichment using a hypergeometric test and then, we carried a permutation analysis (1000 permutations) to determine if the enrichment or depletion seen is more than expected by chance. An enrichment was considered significant if the hypergeometric *p* value was lower than 0.0005 and the permutation *p* value was lower than 0.05.

### MAGMA analysis
The MAGMA[78] gene-set analysis was carried on the significant sex-DE genes (*q* value <0.01) separating the male- and female-biased genes both for prenatal and adult forebrain. The same analysis was also carried out on shared sex-DE genes, and prenatal-specific sex-DE genes (again separating the male- and female-biased genes). Nine disease traits were analyzed: ADHD,[93] Alzheimer,[94] Anorexia,[95] ASD,[96] Bipolar disorder,[97] Epilepsy,[98] Major depressive disorder,[99] Schizophrenia[100] and Tourette syndrome.[101] The MAGMA analysis test whether the genes from the different gene sets are more strongly associated with the phenotypes of interest compared to other genes. We mapped the HapMap3 single nucleotide polymorphisms to protein-coding and lincRNA genes within 50 kb upstream and downstream, using the 1,000 Genomes Phase 3 European dataset as the reference for variant locations and Gencode annotation (version 26) as the reference for gene locations. For all other steps, we followed the standard MAGMA pipeline. We filtered significant enrichment using the *p* value threshold of $6.9 \times 10^{-4}$ (0.05/9 × 8) to account for multiple testing.

### Hypergeometric enrichment analysis

We performed an enrichment analysis on sets of genes linked to neurological or neurodevelopmental disorders through large exome-sequencing studies. Meaning that most genes in these lists were genes where loss-of-function variants were found to be associated with the pathology. The four gene sets are the following: SCZ (Schizophrenia exome meta-analysis consortium,[102] 244 genes), epilepsy[103] (15 genes), neurodevelopmental disorders with epilepsy[104] (33 genes), neurodevelopmental disorder[105] (662 genes), ASD[105] (183 genes), bipolar disorder (Bipolar Exome sequencing project, https://bipex.broadinstitute.org/, 85 genes) and multiple sclerosis (MS)[106–108] (28 genes). We also choose two control gene sets from neurological disorders with older age of onset: Alzheimer[109] (20 genes) and amyotrophic lateral sclerosis (ALS)[110] (36 genes). In addition, we did an enrichment analysis on the gene lists from the MetaBrain co-expression network[66] for which at least 5 genes passed the 5% FDR threshold from the downstreamer analysis. We have gene lists for the 3 following diseases: SCZ, MS and ALS.

For each gene set, we computed the relative enrichment in prenatal and adult sex-DE genes as well as on the prenatal-specific and shared sex-DE genes defined with the Bayesian model. For all these analyses, we separated the genes lists by direction of effect (female- or male-biased). We tested the enrichment using a hypergeometric test. As the sex-DE genes are overall more highly expressed than non-sex-DE ones (Figures S8, S16C, and S16D), we accounted for this confounder (i.e., level of expression) in our enrichment analysis through matching our background set of genes to genes with similar level of expression in the brain. Finally, we carried a permutation analysis to determine if the enrichment seen is more than expected by chance.

We filtered significant enrichment using the hypergeometric test $p$ value with a threshold of $6.9 \times 10^{-4}$ (0.05/9 × 8) and $7.8 \times 10^{-4}$ (0.05/8 × 8) for the enrichment of exome-sequencing studies gene lists and the enrichment of the MetaBrain co-expression network gene lists, respectively, to account for multiple testing. In both cases, we also applied the threshold of 5% on the permutation $p$ value.

### Enrichment analysis of Mulvey et al. 2024[65] gene lists

We performed an enrichment analysis of the prenatal-specific and shared sex-DE in 16 lists of genes linked to neuropsychiatric disorder and 2 control gene sets. These gene sets were compiled from both common-variant (GWAS and TWAS) and rare-variant (whole exome sequencing and DisGeNET) studies in a study by Mulvey and colleagues.[65] We used the same hypergeometric test as for the other disease enrichment analyses accounting for the level of expression (i.e., matching our background set of genes to genes with similar level of expression in the brain). We then carried out a permutation analysis (1000 permutation) to determine if the enrichment was more than expected by chance. We defined a significant enrichment using the hypergeometric test $p$ value with a threshold of $6.9 \times 10^{-4}$ (0.05/18 × 4) to account for multiple testing and a threshold of 0.05 on the permutation $p$ value.

### Enrichment analysis of ASD case-control gene list

We performed an enrichment analysis of 4 gene lists differentially regulated in an ASD case-control study[111]: genes upregulated in ASD patients, genes downregulated in ASD patients, and 2 ASD-associated co-expression network modules (called M12 and M16 in the publication). We used the same hypergeometric test as for the other disease enrichment analyses accounting for the level of expression (i.e., matching our background set of genes to genes with similar levels of expression in the brain). We then carried out a permutation analysis (1000 permutations) to determine if the enrichment was more than expected by chance. We defined a significant enrichment using the hypergeometric test $p$ value with a threshold of $4.2 \times 10^{-3}$ (0.05/4 × 4) to account for multiple testing and a threshold of 0.05 on the permutation $p$-value.

### Replication analysis of prenatal sex-DE genes

To assess the generalizability of our results, we compared our prenatal sex-DE genes to the data from Kissel and colleagues[26] study, which we found is the most similar to our dataset. In this study, the authors analyzed the transcriptomic sex differences in two RNA-seq data of the prenatal brain: UCLA and BrainVar. Technical factors could explain the lower overlap with the UCLA data, as the data is from the whole brain and not specifically from the forebrain, and the RNA-seq library is depleted of ribosomal RNA (Ribo Zero) instead of PolyA+ like in the HDBR data.

