## [Document S2. Transparent peer review records for Benoit-Pilven et al. · Cell Genomics]

Early establishment and life course stability of sex biases in the human brain transcriptome

Clara Benoit-Pilven, Juho V. Asteljoki, Jaakko T. Leinonen, Juha Karjalainen, Mark J. Daly, Taru Tukiainen

Summary

Initial submission: Received : Oct 07, 2024

Scientific editor: Sara Rohban

First round of review: Number of reviewers: 3
Revision invited : Dec 16, 2024
Revision received : Feb 07, 2025

Second round of review: Number of reviewers: 2
Accepted : Apr 30, 2025

Data freely available: YES

Code freely available: YES

This transparent peer review record is not systematically proofread, type-set, or edited. Special characters, formatting, and equations may fail to render properly. Standard procedural text within the editor's letters has been deleted for the sake of brevity, but all official correspondence specific to the manuscript has been preserved.

Referees' reports, first round of review

Reviewer #1:

Benoit-Pilven et al. aim to elucidate the genetic and biological underpinnings of sex differences in brain-related phenotypes by examining transcriptomic sex biases in both developing and adult human forebrains. They identified a significant number of sex-DE genes in the prenatal brain and observed substantial consistency in sex effects between developing and adult brains, with relatively few sex-DE genes unique to the adult forebrain. Their findings suggest that the early testosterone surge may contribute to the establishment of prenatal-specific sex-DE genes, and that genes escaping XCI display consistent sex-DE across life stages. However, limited direct overlap was found between sex-DE genes and genes associated with brain disorders, implying that sex-DE genes may influence disease susceptibility indirectly by modulating gene network activity. The following comments should be addressed to enhance the study's implications for understanding sex differences in brain function and disease susceptibility.

1) While the study found no enrichment of sex-DE genes among brain disorder-associated genes, a comprehensive evaluation was not conducted. The current gene lists from exome sequencing or GWAS may not fully encompass disease-related genes. Genes implicated in brain disorders also arise from differential expression between cases and controls and from gene co-expression networks associated with these disorders. Furthermore, MAGMA only maps SNPs to genes based on location, overlooking critical information like eQTL. Disease-associated genes identified via QTL-based methods (e.g., TWAS, SMR, COLOC) were not included, which may partly explain the lack of overlap with sex-DE genes.

2) In the absence of gene-level enrichment, the study shifts its focus to disease networks and reports significant enrichment in SCZ and MS. Why these two diseases specifically? ASD and MDD exhibit stronger sex biases but do not show enrichment for sex-DE genes. Is the observed enrichment in SCZ and MS due to a larger number of genes involved in their co-regulatory networks? Could the authors provide a negative control, identifying diseases for which sex-DE genes should not be enriched?

3) The authors use ASD as an example to emphasize the importance of studying sex differences in the brain; however, their results do not show enrichment for ASD-related genes or networks, which contrasts with previous findings (10.1038/ncomms10717; 10.1016/j.biopsych.2020.11.023). How do the authors interpret these results, and how do they reconcile the discrepancy with previous studies? As mentioned in Q1, including additional disease-associated genes in the enrichment analysis may help clarify these differences. A more comprehensive analysis is needed rather than a brief discussion at the end of the manuscript.

4) HDBR is a valuable resource for fetal brain samples, and the authors determined sample sex by checking X and Y chromosome reads. However, were any of these samples carrying genetic disabilities (e.g., 1p36 deletion syndrome, 4p deletion syndrome, Trisomy 18), given those are not healthy individuals. Were these conditions balanced between male and female samples? If not, such genetic factors could act as confounding variables influencing the observed sex-DE results.

5) Expanding the sample set to include more post-conception weeks could increase the study's power. For example, Cindy Wen's study analyzed 672 unique brain samples across 4 to 39 post-conception weeks (10.1126/science.adh0829), which could address the limitation in early developmental coverage and better capture hormonal influences.

6) The study uses CIBERSORTx to estimate changes in cell type proportions along development. To extend this analysis, did the authors consider using alternative deconvolution methods to estimate expression levels within each cell type and assess sex-DE at the cell type level? This approach could directly clarify which cell types drive bulk-level sex-DE and identify cell-type-specific enrichment related to testosterone surges and XCI escape.

7) The study finds opposite sex-DE effects between fetal and adult brains. Could the authors propose possible explanations for these opposing directions? Are these results random, or do they reflect interactions with environmental factors? What is the underlying biology?

8) The finding of 74 TFBS in male-biased genes versus only 8 in female-biased genes is noteworthy, as this disparity exceeds the difference observed in the sex-biased genes themselves. Additionally, both androgen and estrogen receptors are enriched in male-biased genes. Why? Author only present these results, but limited explanation or discussion on the findings.

9) The attenuated sex bias in adult brains is intriguing, with the Bayesian model comparison showing only 0.7% of adult-specific sex-DE genes. The authors suggest that intrinsic biological factors, such as sex chromosomes or early hormonal effects, contribute to sex-DE genes. Sounds reasonable, however, critical windows for gonadal hormone differences exist during the third trimester, around birth, and at puberty. Why do hormones during these later stages appear to contribute less to sex-DE than those in very early development? Are these fetal-adult sex-DE differences potentially influenced by technical factors, such as sequencing platforms, depth, or other confounding effects that could attenuate sex-DE in adult brains?

10) The absence of replication in independent cohorts may limit the generalizability of these findings. As noted in Q5, additional brain data would be beneficial to validate the results, particularly those from fetal brains.

Reviewer #2:

The manuscript, "Early establishment and life course stability of sex biases in the human brain transcriptome" by Benoit-Pilven et al. presents analyses of sex-specific differential gene expression in the human brain. The manuscript is generally clear, the analyses are interesting and well-rationalized and the findings will be of broad interest. The authors are careful to discuss limitations of the study and avoid over-stating their findings. One thing that should be considered is to make it more clear at various points in the manuscript which aspects of the data used have been reported on in prior manuscripts. Whether or not the conclusions in this manuscript are in line with or diverge from prior manuscripts would be interesting for the reader to know.

Reviewer #3:

The manuscript presents an interesting analysis of unique HDBR data (i.e., human preterm fetal brain tissue) to examine the effect of transcript abundance for early brain development by sex. The unique resource that the authors analyze left me with many questions as did the choices around using "pseudotime." Given the questions I had around those aspects of the study, it was difficult for me to feel confident that the authors were comparing the same notion across prenatal and adult brain tissue, although it is possible they were. The support for the testosterone surge contributing to the sex differences was not especially compelling to me but these data do not strongly argue against it. The exploration of the sex-DE in disease was interesting although difficult to draw any firm conclusions from.

Major Comments

1. The term pseudotime is not precise, especially in the context used in the work, and it conflates time with sample similarity that could be influenced by things other than time. For these reasons I would strongly encourage not perpetuating its use in the context of donated human tissue if possible.
 - Where the term makes some sense is in the context of cellular models that are getting assayed/harvested at different time points, but it is simply a method of looking at similarity among samples and similarity could result from many factors beyond 'age' of the sample. For instance, technical (e.g., tissue handling) or genetic/environmental (e.g., relatives, infection). The reason that the pseudotime approach may appear more "powered" compared to the traditional developmental stages could also be that it is capturing other sources of variance or inducing colliders since it is derived from the expression data that is being tested. I was very struck by their finding of 31 sex-DE genes by development stages but their finding of 1000s of sex-DE genes using the pseudotime measure, which makes me concerned this is beyond what I'd guess would be seen simply with a boost in power.
 - I recognize that the term is popular and used in the analysis of human tissue samples -- even among high profile journals.
 - I recognize that assigning the age for prenatal tissue is challenging so using the approach to ordering samples is not unreasonable, yet it seems like more variance is being captured by whatever pseudotime is absorbing than simply age but my view (and the authors) is speculation since I do not see how this could be tested. My view is that there is some kind of collider effect being induced since the transcript used for the pseudotime are correlated with other transcripts that are being tested. Exploring how the pseudotime construct is created could help understand this problem.
2. Technically, it is not clear how they account for the repeated measures with the pseudotime analysis or what should be done. It seems like they include all samples, which might be fine. What is the influence of including repeated measures? How are different regions handled? I suspect this is more relevant for GTEx since samples include regions that have considerably different cytoarchitecture (e.g., cortical surface to nuclei). Were they estimated independently for each region and then the average was taken? Also, what is the rationale for selecting the genes? Why 3000? Why not genes with the median variance rather than the extreme? Considering the authors hinge so much of the analysis on using results the decisions around generating this ordering are under explored.
3. The comparison between HDBR and GTEx is difficult to reconcile since it is not clear that the pseudotime metric, which is dependent on the structure of the data, is scaled similarly between the datasets or is otherwise readily comparable, which makes it hard to interpret the comparisons.
4. Sample characteristics are an important issue here, but not clearly presented.
 - The reason for voluntary termination of a pregnancy is broad and includes issues that could easily influence the findings (e.g., chromosomal issues, malformations, or viability) that ought to be the focus

of some sensitivity analysis. GTEx, for instance, was primarily facilitated by organ procurement organizations, which is why you see a male predominance of the donors (due to accidents, which reflects the general trend in US mortality). The extensive genetic characterization and age-range suggest little influence of chromosomal re-arrangements/duplications/deletions that might be present in prenatal samples on findings from GTEx donors. Karyotypes, genotyping, or other measures ought to be explored if possible. Ideally, close relatives should be identified and one randomly dropped since family members are more likely to be similar than individuals in the general population.

5. The claim from the introduction that "reveals a hormonal contribution to the prenatal-specific sex differences" is beyond what I think could be supported. It was possibly "consistent with ..." that interpretation is my suggested phrasing.

Minor Comments

1. It would have helped me if they were clearer that they were using linear mixed model for repeated measures and gave the individuals before samples since the number of individuals is the main driver for power. Whenever the samples are shared the unique individuals ought to be shared first (e.g., in the introduction, results, figures, and discussion).

2. The order of sharing the results was a bit confusing since the sex-interaction analysis is introduced in the "Pseudotime inference for prenatal forebrain samples" section and then mentioned again in the "Widespread sex biases in the prenatal brain" section.

Authors' response to the first round of review

Our replies are highlighted with **blue font** and edits made in the manuscript text in **purple font**.

Reviewer #1:

Benoit-Pilven et al. aim to elucidate the genetic and biological underpinnings of sex differences in brain-related phenotypes by examining transcriptomic sex biases in both developing and adult human forebrains. They identified a significant number of sex-DE genes in the prenatal brain and observed substantial consistency in sex effects between developing and adult brains, with relatively few sex-DE genes unique to the adult forebrain. Their findings suggest that the early testosterone surge may contribute to the establishment of prenatal-specific sex-DE genes, and that genes escaping XCI display consistent sex-DE across life stages. However, limited direct overlap was found between sex-DE genes and genes associated with brain disorders, implying that sex-DE genes may influence disease susceptibility indirectly by modulating gene network activity. The following comments should be addressed to enhance the study's implications for understanding sex differences in brain function and disease susceptibility.

We thank the reviewer for the comprehensive and constructive evaluation of our work.

1) While the study found no enrichment of sex-DE genes among brain disorder-associated genes, a comprehensive evaluation was not conducted. The current gene lists from exome sequencing or GWAS may not fully encompass disease-related genes. Genes implicated in brain disorders also arise from differential expression between cases and controls and from gene co-expression networks associated with these disorders. Furthermore, MAGMA only maps SNPs to genes based on location, overlooking critical information like eQTL. Disease-associated genes identified via QTL-based methods (e.g., TWAS, SMR, COLOC) were not included, which may partly explain the lack of overlap with sex-DE genes.

This is an excellent remark from the reviewer. We initially focused our disease enrichment analysis specifically on genes identified in genetic studies as these represent more confidently the etiologial processes of brain-related diseases, and as such we reason these are particularly relevant genes to examine to understand if and how sex-DE can impact the development or risk for disease. Genes that are DE between cases and controls, which are more typically used in assessing overlap with sex-DE, can be informative for disease-related processes but do not necessarily similarly reflect processes causal for disease. We understand that such a specific focus on genes from genetic studies may have been a source of confusion and raised concerns on whether we detect similar overlaps as reported in previous studies.

We have now conducted an enrichment analysis of neuropsychiatric disorders analyzed in the following preprint Mulvey et al., 2024¹. The lists of genes include both genes reported in common-variant (GWAS and TWAS) and rare-variant (whole exome sequencing and DisGeNET) studies. For this enrichment analysis, we used the same method we previously used in our manuscript, i.e. hypergeometric test and permutation test controlling for the gene level of expression (see Methods section). We did not see any significant enrichments for any neuropsychiatric disorder in our prenatal-specific and shared sex-DE gene lists (see Figure a below).

Figure a: Enrichment analysis of 16 neuropsychiatric disorders and 2 control gene sets in prenatal specific and shared sex-DE genes (separated by direction of effect, i.e. female-biased and male-biased genes). The violinplots represent the distribution of the relative enrichment for the 1000 random gene lists and the point the enrichment for the true gene list.

These results differ from the Mulvey et al., 2024¹ results where they found an enrichment of male-biased genes for several neuropsychiatric disorders, such as autism spectrum disorder, neurodevelopmental disorder, and schizophrenia. These discordant results likely stem from the difference in the forebrain region analysed. While Mulvey at al., 2024¹ focused on two very specific sub-regions of the forebrain (ventromedial hypothalamus and arcuate) with the help of spatial

transcriptomic technology, we analysed the forebrain as a whole and as such may miss cell-type dependent effects.

We now mention these results in the results section “Genes with shared sex bias associate with disease networks” (page 19 lines 530-536). We also describe this new enrichment analysis in the supplementary text (**Document S1**) and we have added a supplementary figure (**Figure S19**).

2) In the absence of gene-level enrichment, the study shifts its focus to disease networks and reports significant enrichment in SCZ and MS. Why these two diseases specifically? ASD and MDD exhibit stronger sex biases but do not show enrichment for sex-DE genes. Is the observed enrichment in SCZ and MS due to a larger number of genes involved in their co-regulatory networks? Could the authors provide a negative control, identifying diseases for which sex-DE genes should not be enriched?

We thank the reviewer for the pertinent comment. For the network enrichment analysis, we used the available data from the MetaBrain co-regulation network study² where the analysis was carried out on 5 brain disorders: SCZ, Parkinson disease, MS, Alzheimer disease, and amyotrophic lateral sclerosis. Therefore, unfortunately, data for ASD and MDD were not available and the enrichment for these diseases were only tested using genes confidently linked to the etiological process (see reply for comment #1). We also want to emphasize that we performed a permutation test for the enrichment analyses to minimize the impact of the size of the gene lists on the results.

We have now clarified the availability of the co-regulation network data for these 5 brain disorders in the results section “Genes with shared sex bias associate with disease networks” (page 20 lines 550-551).

3) The authors use ASD as an example to emphasize the importance of studying sex differences in the brain; however, their results do not show enrichment for ASD-related genes or networks, which contrasts with previous findings (10.1038/ncomms10717; 10.1016/j.biopsych.2020.11.023). How do the authors interpret these results, and how do they reconcile the discrepancy with previous studies? As mentioned in Q1, including additional disease-associated genes in the enrichment analysis may help clarify these differences. A more comprehensive analysis is needed rather than a brief discussion at the end of the manuscript.

We want to thank the reviewer for the very relevant comment. First, we want to emphasize that our results do not, in fact, conflict with previous findings on ASD as the studied gene sets are derived from different sources, i.e., we mainly focused on genes involved in the etiology of the disease while previous studies have primarily investigated enrichments for genes discovered in case-control comparisons.

To demonstrate this, we have now compared our sex-DE genes (prenatal-specific and shared) to the genes differentially expressed between cases and controls for ASD³ (see Figure b below). We observed similar enrichment to those reported in the Werling et al., 2016, ASD study⁴. In brief, the shared female-biased genes were enriched in genes downregulated in ASD cases as well as the ASD-associated co-expression module (M16) enriched for genes involved in immune and inflammatory responses. The shared male-biased genes were enriched for the co-expression module (M12) downregulated in ASD

and enriched for genes with neuronal and synaptic functions. These enrichments were specific to shared genes and no enrichment of the prenatal-specific sex-DE genes was observed.

Figure b: Enrichment analysis of case-control ASD associated genes in prenatal specific and shared sex-DE genes (separated by direction of effect, i.e. female-biased and male-biased genes). The violinplots represent the distribution of the relative enrichment for the 1000 random gene lists and the point the enrichment for the true gene list.

We have now included these new results in the results section “Genes with shared sex bias associate with disease networks” (page 19 lines 537-541). We also describe the analysis done in the supplementary text (**Document S1**) and we have added a supplementary figure (**Figure S20**).

4) HDBR is a valuable resource for fetal brain samples, and the authors determined sample sex by checking X and Y chromosome reads. However, were any of these samples carrying genetic disabilities (e.g., 1p36 deletion syndrome, 4p deletion syndrome, Trisomy 18), given those are not healthy individuals. Were these conditions balanced between male and female samples? If not, such genetic factors could act as confounding variables influencing the observed sex-DE results.

We apologize for not describing the HDBR sample characteristics in detail in the manuscript in the first place and thank the reviewer for pointing that out.

The majority of the HDBR samples have been karyotyped and only samples with normal karyotype material are provided for research (<https://www.hdbr.org/>). The HDBR Expression set⁵ that we are analysing is a subset of these samples declared as karyotypically normal. Overall, only a small fraction of the samples (4%, in 2015⁶) have chromosomal abnormalities (most commonly trisomy 21 and monosomy X). Further, the 9% of samples with abnormal phenotypes⁶ constitute a separate HDBR cohort (<https://hdbratlas.org/abnormal-exome-seq-data.html>), suggesting that fetal abnormalities

explain only some of the abortions through which the HDBR material is obtained. While karyotyping may not detect smaller chromosomal aberrations, such as the 1p36 deletion syndrome, these fetuses may, nevertheless, show phenotypic signs that would have resulted in their exclusion. Given these details, we are quite confident that we are largely analysing samples representing normal fetal development and the related sex differences.

We have now clarified the status of the samples as karyotypically normal in the “Description of datasets” section of the main text (page 6 line 110) and in the methods (page 33 lines 957-959) and added relevant citations describing the HDBR data.

5) Expanding the sample set to include more post-conception weeks could increase the study's power. For example, Cindy Wen's study analyzed 672 unique brain samples across 4 to 39 post-conception weeks (10.1126/science.adh0829), which could address the limitation in early developmental coverage and better capture hormonal influences.

We thank the reviewer for this excellent suggestion. We agree that expanding the sample size and the range of post-conception weeks of the analyzed data could increase the study's power. Prompted by the reviewer's comment, we explored the possibility to validate our findings in other data sets.

The data used in Wen et al., 2014⁷ is from 5 different publications and we find this would introduce substantial heterogeneity in the sex-DE analysis. For example, a large number of samples are not from the forebrain, i.e., the region we focus on, but from the whole brain or the hindbrain. Of note, one of these 5 datasets (representing 163 individuals out of the 654 individuals, ~22%) is the one we are currently analysing in our manuscript and most of the early first trimester samples came from this dataset. Two other datasets, BrainVar⁸ (116 individuals, ~84% coming from the 2nd trimester) and UCLA⁹ (211 individuals from the second trimester only), included in the Wen et al study were used to assess prenatal transcriptomic sex differences in a recent Kissel et al., 2024 study¹⁰. We have now taken advantage of these sex-DE results from Kissel et al paper to look at the replicability of our sex-DE results (see response to comment #10). Confirming the difficulty of merging different datasets to analyse sex differences, Kissel and colleagues analyzed the two datasets separately and then did a meta-analysis. Given these above points, we do not think that using the Wen et al dataset would, unfortunately, address the limited coverage of early development or help understand the hormonal influence.

We have now added to the revised manuscript the replication analyses with the Kissel et al data as stated in more detail in response to comment #10.

6) The study uses CIBERSORTx to estimate changes in cell type proportions along development. To extend this analysis, did the authors consider using alternative deconvolution methods to estimate expression levels within each cell type and assess sex-DE at the cell type level? This approach could directly clarify which cell types drive bulk-level sex-DE and identify cell-type-specific enrichment related to testosterone surges and XCI escape.

This is a very interesting point from the reviewer. We have now analyzed the prenatal dataset using the TOAST method¹¹ to study sex-DE at the cell type level using a model that includes sex and pseudotime as

covariates in addition to the cell type. As recommended by the authors of this method, we also did a permutation analysis to identify “real” effects.

After filtering the significant sex-DE genes per cell type (FDR<0.05 and permutation p-value<0.05), we observed that the neuronal progenitor cells (NPC) and the excitatory neurons are the cell types with the most sex-DE effects (2697 and 330 respectively, after removing the genes used to define the sex of the samples). The large overlap between the NPC-specific sex-DE genes and the prenatal sex-DE genes (N=629, 19.74% of the prenatal sex-DE genes) shows that the NPC partly drives the prenatal bulk-level sex-DE.

We did not observe an enrichment for any cell type related to escape from XCI, in agreement with the idea that this is a process shared by all cell-types. For instance, in NPC, 16/69 escape genes were sex-DE (hypergeometric enrichment p-value = 0.06). We did a similar enrichment analysis for androgen-responsive genes and found a suggestive enrichment of testosterone-responsive genes in excitatory neurons specific sex-DE genes (hypergeometric enrichment p-value = 0.01).

This analysis is now mentioned in section “Shared sex differences in prenatal and adult brain reflect neuronal functions” (page 15 lines 387-390) of the results and described in more detail in the supplementary text (**Document S1**).

7) The study finds opposite sex-DE effects between fetal and adult brains. Could the authors propose possible explanations for these opposing directions? Are these results random, or do they reflect interactions with environmental factors? What is the underlying biology?

We indeed find some genes that show opposite signs of sex-DE between the prenatal and adult brain. These genes, 45 in total, are a minority compared to the general pattern of sex-DE (719 genes with similar direction of sex-DE), but are an interesting group of genes to look into more closely.

We did not find specific characteristics explaining these sex-DE genes with opposite effects. For instance, these 45 genes are as tissue-specific as the shared sex-DE genes and their mean expression level is similar to the prenatal-specific sex-DE genes in all cell types. As such, it is likely that there are diverse and mixed reasons as to why these 45 genes display opposite prenatal and adult sex-DE effects. However, we found some interesting example genes that are relevant for brain development such as SCN8A and KCNB1. KCNB1 is a gene encoding for a part of a potassium channel called Kv2.1. It has been shown to be highly expressed in neocortical pyramidal cells¹² and mutations in this gene have been linked to encephalopathy, epilepsy, and developmental delay¹³.

This KCNB1 gene is now briefly described in section “Shared sex differences in prenatal and adult brain reflect neuronal functions” (page 14 lines 349-352) and was added as an example in **Figure 2E**.

8) The finding of 74 TFBS in male-biased genes versus only 8 in female-biased genes is noteworthy, as this disparity exceeds the difference observed in the sex-biased genes themselves. Additionally, both androgen and estrogen receptors are enriched in male-biased genes. Why? Author only present these results, but limited explanation or discussion on the findings.

We thank the reviewer for pointing that out as these are, indeed, quite intriguing findings. The hormonal regulation in early brain development is complex. Importantly, AR, that mediates the effects

of androgens on gene expression, has an auto-regulation feedback loop in the presence of androgens, meaning that its expression is downregulated upon androgen binding. This could explain some of the seemingly paradoxical findings, such as AR expression being upregulated in female fetuses compared to males and prenatal-specific male-biased genes being enriched among those downregulated by androgen treatment. We have now included more discussion on the complexity of the hormonal regulation in early brain development as well as on these seemingly paradoxical findings (see Discussion page 23 lines 645-657).

9) The attenuated sex bias in adult brains is intriguing, with the Bayesian model comparison showing only 0.7% of adult-specific sex-DE genes. The authors suggest that intrinsic biological factors, such as sex chromosomes or early hormonal effects, contribute to sex-DE genes. Sounds reasonable, however, critical windows for gonadal hormone differences exist during the third trimester, around birth, and at puberty. Why do hormones during these later stages appear to contribute less to sex-DE than those in very early development? Are these fetal-adult sex-DE differences potentially influenced by technical factors, such as sequencing platforms, depth, or other confounding effects that could attenuate sex-DE in adult brains?

This is a finding that was also intriguing to us. The limited current availability of data from the different stages of human brain development makes the dissection of the impact of the different critical developmental windows challenging. Projects like the developmental GTEx¹⁴ will hopefully help in closing this important gap in our knowledge. In the revised manuscript, we have added a sentence in the Discussion (pages 24 lines 690-693) to discuss this aspect.

We acknowledge the potential influence of technical factors and we did our best to account for those by using surrogate variables (svas). Svas are an efficient way to capture technical and other variations unrelated to the variable of interest in the DE analyses when these technical factors are not directly available¹⁵.

10) The absence of replication in independent cohorts may limit the generalizability of these findings. As noted in Q5, additional brain data would be beneficial to validate the results, particularly those from fetal brains.

We understand the concern about the generalizability of our findings and apologize for omitting a replication analysis. First, we wish to highlight that an exact validation of our findings can be challenging, as the other prenatal brain data sets that have been analysed for sex-DE are typically from a slightly different age range, can be from a different brain region, and the methods applied, including covariate adjustments, are not necessarily comparable.

Nevertheless, to ensure our findings are generalizable, we have now compared our prenatal sex-DE results to the two sets of sex-DE genes (BrainVar and UCLA) from the Kissel et al., 2024 study¹⁰, which we found are the most similar to our data set and approach. In both analyses, the overlap of the sex-DE genes was large (87.18% and 68.66% of the significant sex-DE genes from BrainVar and UCLA respectively were also significant in our analysis). Technical factors could explain the lower overlap with the UCLA data, as the data is from the whole brain and not specifically from the forebrain, and the RNA-seq library is depleted of ribosomal RNA (Ribo Zero) instead of PolyA+ like in the HDBR data.

These analyses show that our results capture similar sex-DE signatures as reported previously, therefore supporting that these findings can be generalizable to other prenatal brain data sets.

We have now added these results in the section “Widespread sex biases in the prenatal brain” (page 9 lines 213-217) and in the supplementary text (**Document S1**).

Reviewer #2:

The manuscript, "Early establishment and life course stability of sex biases in the human brain transcriptome" by Benoit-Pilven et al. presents analyses of sex-specific differential gene expression in the human brain. The manuscript is generally clear, the analyses are interesting and well-rationalized and the findings will be of broad interest. The authors are careful to discuss limitations of the study and avoid over-stating their findings. One thing that should be considered is to make it more clear at various points in the manuscript which aspects of the data used have been reported on in prior manuscripts. Whether or not the conclusions in this manuscript are in line with or diverge from prior manuscripts would be interesting for the reader to know.

We thank the reviewer for the positive and constructive feedback.

We have now included in the revised manuscript more comparisons with previous findings as listed below.

We replicated the ASD-related genes enrichment from the Werling et al., 2016 study⁴ (see response to reviewer #1 comment #3). However, to our knowledge, this result has not been dissected before into genes implicated in the disease etiology, as we do in our manuscript.

We used the neuropsychiatric disorder gene lists compiled in the Mulvey et al., 2024 study¹ to explore further the disease enrichments of our sex-DE genes. In agreement with our original findings, we didn't observed any enrichment of these gene lists in our prenatal-specific and shared sex-DE gene lists (see response to reviewer #1 comment #1).

We also used two datasets from the Kissel et al., 2024 study¹⁰ and showed a large overlap of their prenatal sex-DE genes with ours (see response to reviewer #1 comment #10).

Reviewer #3:

The manuscript presents an interesting analysis of unique HDBR data (i.e., human preterm fetal brain tissue) to examine the the effect of transcript abundance for early brain development by sex. The unique resource that the authors analyze left me with many questions as did the choices around using "pseudotime." Given the questions I had around those aspects of the study, it was difficult for me to feel confident that the authors were comparing the same notion across prenatal and adult brain tissue, although it is possible they were. The support for the testosterone surge contributing to the sex differences was not especially compelling to me but these data do not strongly argue against it. The exploration of the sex-DE in disease was interesting although difficult to draw any firm conclusions from.

We want to thank the reviewer for the critical evaluation of our work and the many excellent points raised. To accommodate the comments from the reviewer, we have now extensively explored the characteristics and behavior of the pseudotime variable, and introduced multiple changes to the manuscript prompted by the feedback and the new analyses we have conducted, as detailed in the

responses to the specific comments below. Overall, based on the new analyses, we still find the use of pseudotime justified and find that our results from the sex-DE analyses and the comparison of sex effects between HDBR and GTEx is valid. However, we now highlight the limitations of the approach more carefully.

For instance, we have changed the wording in the manuscript to avoid the interpretation of pseudotime only as a continuous proxy for developmental stage but rather highlight it as a linear description of gene expression variability across the samples that in part reflects prenatal brain development. We added some of the new analyses in section “Pseudotime inference for prenatal forebrain samples” of the main text (pages 7-8 lines 150-168 and 174-186) and in section “Attenuated sex bias in the adult brain” (page 11 lines 264-276) along with new supplemental figures (**Figures S6, Figure S7, Figure S11**) and present them in full in **Document S1**. Also, we have added text in the discussion section (page 25 lines 705-709) to describe the challenges associated with the use and interpretation of pseudotime.

Major Comments

1. The term pseudotime is not precise, especially in the context used in the work, and it conflates time with sample similarity that could be influenced by things other than time. For these reasons I would strongly encourage not perpetuating its use in the context of donated human tissue if possible.

- Where the term makes some sense is in the context of cellular models that are getting assayed/harvested at different time points, but it is simply a method of looking at similarity among samples and similarity could result from many factors beyond 'age' of the sample. For instance, technical (e.g., tissue handling) or genetic/environmental (e.g., relatives, infection). The reason that the pseudotime approach may appear more "powered" compared to the traditional developmental stages could also be that it is capturing other sources of variance or inducing colliders since it is derived from the expression data that is being tested. I was very struck by their finding of 31 sex-DE genes by development stages but their finding of 1000s of sex-DE genes using the pseudotime measure, which makes me concerned this is beyond what I'd guess would be seen simply with a boost in power.

- I recognize that the term is popular and used in the analysis of human tissue samples -- even among high profile journals.

- I recognize that assigning the age for prenatal tissue is challenging so using the approach to ordering samples is not unreasonable, yet it seems like more variance is being captured by whatever pseudotime is absorbing than simply age but my view (and the authors) is speculation since I do not see how this could be tested. My view is that there is some kind of collider effect being induced since the transcript used for the pseudotime are correlated with other transcripts that are being tested. Exploring how the pseudotime construct is created could help understand this problem.

We understand that pseudotime inference is not necessarily a standard approach to be applied in a setting like ours, although it has been extensively used to model developmental trajectories in diverse organisms. Hence, we acknowledge that we should have investigated the underlying characteristics in more detail and provided more evidence to support the validity of the approach. We hope the following details and new results will help address the reviewer's concerns.

The reviewer was struck by the large difference in significant sex-DE genes between the analyses using categorical developmental stages and the pseudotime-adjusted analysis. We are afraid that the comparison of the sex-DE results only at the level of significant genes we initially included in the manuscript may have given a biased impression of the degree of differences between the two analyses. Using the strict threshold of $q\text{-value} < 0.01$, we indeed find only 31 sex-biased genes in the analysis where the developmental stage is included as a categorical covariate (sex-DE_devstage), which is in great contrast to the 3187 sex-DE genes identified in the analysis with the pseudotime covariate (sex-DE_pseudo). A more in-depth analysis, however, shows there are much greater similarities between the two analyses. We, for instance, see a high consistency in both sex-DE effect directions (87% consistency for the sex-DE genes from the sex-DE_pseudo analysis, permuted $p\text{-value} < 0.001$) and effect estimates (Pearson correlation across autosomal sex-DE genes from sex-DE_pseudo $r = 0.790$, $p\text{-value} < 1 \times 10^{-10}$). Further, the π_1 statistic indicates that a significant fraction of the sex-DE genes from sex-DE_pseudo show non-null p -values in sex-DE_devstage ($\pi_1 = 0.42$).

We additionally assessed how much of the apparent power gain with pseudotime might be related to the use of a continuous covariate in the place of the categorical stages. To this end, we divided the samples into 8 bins of similar size to the developmental stages based on the inferred pseudotime and performed a sex-DE analysis (sex-DE_pseudoBins). Similarly to the sex-DE_devstage analysis, sex-DE_pseudoBins yields a much smaller number of significant sex-DE genes ($N = 18$) than the sex-DE_pseudo, demonstrating the added value of a continuous covariate in the DE analysis. Reassuringly, the sex-DE_pseudoBins results are overall in good agreement with the results from sex-DE_pseudo (89.5% consistency for the sex-DE genes from the sex-DE_pseudo analysis, permuted $p\text{-value} < 0.001$; Pearson correlation across autosomal sex-DE genes from sex-DE_pseudo: $r = 0.765$, $p\text{-value} < 1 \times 10^{-10}$; $\pi_1 = 0.47$ for sex-DE genes from sex-DE_pseudo in sex-DE_pseudoBins).

Given these results, we find that the sex-DE signal is very similar between the two analyses (sex-DE_pseudo and sex-DE_devstage) but the use of continuous pseudotime in the place of the categorical developmental stages does, in fact, reduce noise allowing us to better detect sex-DE signal.

The package we used in the pseudotime inference, phenopath, allows the inclusion of covariates that will be independent of the pseudotime estimates. We, therefore, tested how the sex-DE discovery changes when adjusting the analysis with a pseudotime inferred with the developmental stages defined by HDBR as a covariate (also including sex as a covariate as in the original pseudotime inference). In other words, in this case, pseudotime provides an ordering of the samples that should not reflect the developmental process but may correlate with other biological or technical factors in the data. As expected, the correlation of this new pseudotime variable with the developmental stages is much lower than with the original pseudotime ($\tau = 0.17$ vs $\tau = 0.46$ in the original analysis). However, the two pseudotime estimates are still correlated ($r = 0.76$), suggesting the pseudotime inference, to some degree, also captures other structure in the data (see below and our next reply). The new pseudotime does not, however, provide much added value to the sex-DE analysis. The number of sex-DE genes

(N=52 with q -value<0.01) is marginally higher than in the sex-DE_devstage analysis (N=31) but remains much below the original sex-DE_pseudo analysis (N=3187).

As such, it appears that the use of pseudotime is not a direct correlate of the large number of sex-DE genes. Rather these findings suggest that the boost in the discovery of sex-DE genes is largely related to the use of pseudotime that at least partly captures the developmental trajectory.

To understand if pseudotime is absorbing technical variability in the data, we investigated the correlation of pseudotime with a few technical covariates in HDBR and GTEx. In HDBR, we see a correlation between pseudotime and library size (proxied by total read count) ($r=0.17$, p -value=0.006), but no correlation with time in transit and pseudotime ($r=0.06$, p -value=0.40). In GTEx, we see a clear correlation between pseudotime and RNA integrity number (RIN, $r=-0.46$, p -value= 5.0×10^{-87}). However, the impact of RIN on the pseudotime inference appears minimal, as a pseudotime inferred with RIN as a covariate is highly similar to the original GTEx pseudotime ($r=0.97$, p -value< 2.2×10^{-16}). Other technical covariates, batch and sample ischemic time, were not correlated with pseudotime.

Therefore, pseudotime does, indeed, to a small degree capture technical characteristics of the data.

Multiple changes to text and supplement, see details above in the first comment to the reviewer.

2. Technically, it is not clear how they account for the repeated measures with the pseudotime analysis or what should be done. It seems like they include all samples, which might be fine. What is the influence of including repeated measures? How are different regions handled? I suspect this is more relevant for GTEx since samples include regions that have considerably different cytoarchitecture (e.g., cortical surface to nuclei). Were they estimated independently for each region and then the average was taken? Also, what is the rationale for selecting the genes? Why 3000? Why not genes with the median variance rather than the extreme? Considering the authors hinge so much of the analysis on using results the decisions around generating this ordering are under explored.

It is, indeed, important to clarify how we accounted for the repeated measurements per individual in the GTEx and HDBR data sets. We computed the pseudotime originally for each sample separately, i.e., we treated samples from each individual as independent samples, as there was no option of including random effect variables in the phenopath model. To confirm this approach provides us with valid pseudotime values, we checked that the inferred pseudotime of the different samples from the same individual was more consistent than between random samples (results described in section “Pseudotime inference for prenatal forebrain samples” of the manuscript and **Figure S5B**). To further investigate how the presence of repeated measures affects the pseudotime inference, we have now inferred the pseudotime with the same 2000 most variable genes selecting only 1 sample per individual. This new pseudotime (pseudotime_1sample) shows a very high correlation ($r=0.82$) with our original pseudotime for the selected samples as well as a very good correlation with the developmental stages ($\tau=0.67$). Additionally, we computed a pseudotime taking an average of all samples available per individual and confirmed this pseudotime (pseudotime_mean) shows a high correlation ($r=0.78$) with the original pseudotime.

Overall, these results suggest the approach chosen for pseudotime inference can provide similar per-individual pseudotime estimates irrespective of the exact selection of and correlation structure between the input samples.

In support of the reviewer's thinking that cytoarchitecture might play a role in the pseudotime inference, we noted that pseudotime_1sample shows less correlation with the estimated cell type proportions than the original pseudotime. For instance, the correlation of the pseudotime with the proportion of neuronal progenitor cells is -0.72 with pseudotime_1sample and -0.83 with the original pseudotime. To explore the connection between pseudotime and cytoarchitecture in more detail, we assessed if pseudotime differs between the different forebrain regions captured in GTEx. Confirming that pseudotime partly reflects cell-type dependent expression patterns, we noted some differences in the pseudotime estimates between samples from different brain regions (see Figure c below). For instance, samples from frontal cortex typically received pseudotime estimates smaller than those from hippocampus within an individual (paired Wilcoxon test p -value= 2.8×10^{-25})

Figure c: Boxplot of pseudotime per forebrain region in GTEx data.

These findings suggest that part of the boost in power from the use of pseudotime likely comes from the fact that with this approach we are able to account for some of the between-sample differences within an individual while a categorical developmental stage assigns a single value for all samples per individual.

The reviewer also raises a good question on the selection of input genes for the pseudotime inference. Our original choice of using 2000 most variable genes (not 3000 genes like we had mistakenly written in the first version of the manuscript, our apologies) was driven by the recommendations by the phenopath package developers. We also find that using the most variable genes is a justified choice for the input as a principal component analysis based on these genes captures the developmental differences between the samples (**Figure S2D**).

To better understand how pseudotime inference is impacted by the input, we tested how changing the number of input genes (from 100 to 3000 most variable genes) in the pseudotime analysis changes the inferred pseudotime in HDBR (see Figure d below). We find that reducing the number of input genes, even down to 700 most variable genes, maintains the estimated pseudotime values very consistent with our original pseudotime inferred using the 2000 most variable genes (Pearson's $r > 0.98$). Accordingly, these pseudotime estimates also show a good correlation with the developmental stages (Kendall's tau > 0.41), and across the iterations, the highest correlations are observed between pseudotimes inferred with 1000-2000 input genes. A smaller number of input genes, however, starts to introduce more variability in the pseudotime and also drops correlation with the developmental stages dramatically, suggesting these genes capture a more variable presentation of the samples. With a larger number of input genes (>2000), the correlation with both the original pseudotime and the developmental stages remains generally high, but are, nevertheless, lower than with the 1000-2000 input genes.

Figure d : On the left panel, Pearson's correlation between the original pseudotime and the new inferred pseudotime with a number of input genes ranging from 100 to 3000. On the right panel, Kendall's tau values between the developmental stages and the new inferred pseudotime with the same range of input genes.

Given the drastic change in the correlation of pseudotime and developmental stages between the pseudotime inferred with 601 or 700 most variable genes in the data set, we reason that these 100 genes are most informative for understanding the processes captured by the pseudotime metric that reflect the developmental trajectory. Supporting this idea, we find that in GO enrichment analysis these 100 genes are enriched, e.g., in processes related to development and neurogenesis (such as "regulation

of nervous system development” or “regulation of neurogenesis”), and this set of genes include, for instance, SOX5 and RELN that are known to play roles in brain development^{16,17}.

Overall, these new results demonstrate the robust behavior of the inferred pseudotime irrespective of the exact number of input genes.

Multiple changes to text and supplement, see details above in the first comment to the reviewer.

3. The comparison between HDBR and GTEx is difficult to reconcile since it is not clear that the pseudotime metric, which is dependent on the structure of the data, is scaled similarly between the datasets or is otherwise readily comparable, which makes it hard to interpret the comparisons.

We are afraid we are not quite following the reviewer’s line of thought here. We agree that the pseudotime inferred from GTEx is based on different input genes and at least partly reflects other processes than the pseudotime from HDBR. We, nevertheless, do not see a reason, for the purposes of comparing results from two sex-DE analyses, we should expect the scale of a given covariate to be similar between the two data sets.

To clarify, in our work, we are not comparing pseudotime-DE genes or their effect sizes between HDBR and GTEx. Rather our focus is on the sex-DE genes and their effect sizes and these results (linear regression effect estimates for sex) are not affected by the scaling of a covariate. As such, we consider our approach of comparing HDBR and GTEx sex-DE results valid.

4. Sample characteristics are an important issue here, but not clearly presented.

- The reason for voluntary termination of a pregnancy is broad and includes issues that could easily influence the findings (e.g., chromosomal issues, malformations, or viability) that ought to be the focus of some sensitivity analysis. GTEx, for instance, was primarily facilitated by organ procurement organizations, which is why you see a male predominance of the donors (due to accidents, which reflects the general trend in US mortality). The extensive genetic characterization and age-range suggest little influence of chromosomal re-arrangements/duplications/deletions that might be present in prenatal samples on findings from GTEx donors. Karyotypes, genotyping, or other measures ought to be explored if possible. Ideally, close relatives should be identified and one randomly dropped since family members are more likely to be similar than individuals in the general population.

We apologize for not describing the HDBR sample characteristics in detail in the manuscript in the first place and thank the reviewer for pointing that out.

In the response to comment #4 of Reviewer #1, we explain that the HDBR samples have been cytogenetically karyotyped and only normal karyotyped material is provided for research.

As we are analysing only the gene expression level data, we cannot fully exclude the possibility of a degree of sample relatedness in the HDBR data set. We, however, find the presence of close relatives among the 72 individuals studied quite unlikely. The SNP array data from the HDBR samples that has been analysed in a few previous publications¹⁸ has not indicated any relatedness among the individuals. Further, the HDBR expression samples have been collected across several collection sites in the UK⁵, which limits the chances of related individuals contributing material to the resource.

We have now clarified the status of the samples as karyotypically normal in the “Description of datasets” section of the main text (page 6 lines 110) and in the methods (page 33 lines 943-945) and added relevant citations describing the HDBR data.

5. The claim from the introduction that “reveals a hormonal contribution to the prenatal-specific sex differences” is beyond what I think could be supported. It was possibly “consistent with ...” that interpretation is my suggested phrasing.

We thank the reviewer for their suggestion and we have modified the text accordingly (page 4 line 90).

Minor Comments

1. It would have helped me if they were clearer that they were using linear mixed model for repeated measures and gave the individuals before samples since the number of individuals is the main driver for power. Whenever the samples are shared the unique individuals ought to be shared first (e.g., in the introduction, results, figures, and discussion).

We thank the reviewer for pointing out we had not included this detail in the manuscript. We have added the number of individuals in the text (page 9 line 193-194; page 11 line 257) and figures (**Figure 1** and **Figure S9**) in addition to the number of samples.

2. The order of sharing the results was a bit confusing since the sex-interaction analysis is introduced in the “Pseudotime inference for prenatal forebrain samples” section and then mentioned again in the “Widespread sex biases in the prenatal brain” section.

We thank the reviewer for calling our attention to this confusing part of the results. To make the flow of the results clearer, we have removed the following sentence “Echoing this, we observed that using the developmental stages as a categorical variable in the DE analysis limits the power to detect sex-DE genes (only 31 sex-DE genes with $q\text{-value} < 0.01$; Figure S4A-B) likely due to the low number of samples (2-39) per PCW for each sex.” from the “Pseudotime inference for prenatal forebrain samples” section, and only start presenting sex-DE results in the “Widespread sex biases in the prenatal brain” section (page 9).

References:

1. Mulvey, B., Wang, Y., Divecha, H.R., Bach, S.V., Montgomery, K.D., Cinquemani, S., Chandra, A., Du, Y., Miller, R.A., Kleinman, J.E., et al. (2024). Spatially-resolved molecular sex differences at single cell resolution in the adult human hypothalamus. *bioRxiv*, 2024.12.07.627362. <https://doi.org/10.1101/2024.12.07.627362>.
2. Klein, N. de, Tsai, E.A., Vochteloo, M., Baird, D., Huang, Y., Chen, C.Y., Dam, S. van, Oelen, R., Deelen, P., Bakker, O.B., et al. (2023). Brain expression quantitative trait locus and network analyses reveal downstream effects and putative drivers for brain-related diseases. *Nature Genetics* 55, 377–388. <https://doi.org/10.1038/s41588-023-01300-6>.
3. Voineagu, I., Wang, X., Johnston, P., Lowe, J.K., Tian, Y., Horvath, S., Mill, J., Cantor, R., Blencowe, B.J., and Geschwind, D.H. (2011). Transcriptomic Analysis of Autistic Brain Reveals Convergent Molecular Pathology. *Nature* 474, 380–384. <https://doi.org/10.1038/nature10110>.
4. Werling, D.M., Parikshak, N.N., and Geschwind, D.H. (2016). Gene expression in human brain implicates sexually dimorphic pathways in autism spectrum disorders. *Nature Communications* 7. <https://doi.org/10.1038/ncomms10717>.
5. Lindsay, S.J., Xu, Y., Lisgo, S.N., Harkin, L.F., Copp, A.J., Gerrelli, D., Clowry, G.J., Talbot, A., Keogh, M.J., Coxhead, J., et al. (2016). HDBR Expression: A Unique Resource for Global and Individual Gene Expression Studies during Early Human Brain Development. *Frontiers in Neuroanatomy* 10, 86. <https://doi.org/10.3389/fnana.2016.00086>.
6. Gerrelli, D., Lisgo, S., Copp, A.J., and Lindsay, S. (2015). Enabling research with human embryonic and fetal tissue resources. *Development* 142, 3073–3076. <https://doi.org/10.1242/dev.122820>.
7. Wen, C., Margolis, M., Dai, R., Zhang, P., Przytycki, P.F., Vo, D.D., Bhattacharya, A., Matoba, N., Tang, M., Jiao, C., et al. (2024). Cross-ancestry atlas of gene, isoform, and splicing regulation in the developing human brain. *Science* 384, eadh0829. <https://doi.org/10.1126/science.adh0829>.
8. Werling, D.M., Pochareddy, S., Choi, J., An, J.-Y., Sheppard, B., Peng, M., Li, Z., Dastmalchi, C., Santpere, G., Sousa, A.M.M., et al. (2020). Whole-Genome and RNA Sequencing Reveal Variation and Transcriptomic Coordination in the Developing Human Prefrontal Cortex. *Cell Reports* 31, 107489. <https://doi.org/10.1016/j.celrep.2020.03.053>.
9. Walker, R.L., Ramaswami, G., Hartl, C., Mancuso, N., Gandal, M.J., de la Torre-Ubieta, L., Pasaniuc, B., Stein, J.L., and Geschwind, D.H. (2019). Genetic Control of Expression and Splicing in Developing Human Brain Informs Disease Mechanisms. *Cell* 179, 750–771.e22. <https://doi.org/10.1016/j.cell.2019.09.021>.
10. Kissel, L.T., Pochareddy, S., An, J.-Y., Sestan, N., Sanders, S.J., Wang, X., and Werling, D.M. (2024). Sex-Differential Gene Expression in Developing Human Cortex and Its Intersection With Autism Risk Pathways. *Biological Psychiatry Global Open Science* 4, 100321. <https://doi.org/10.1016/j.bpsgos.2024.100321>.
11. Li, Z., Wu, Z., Jin, P., and Wu, H. (2019). Dissecting differential signals in high-throughput data from complex tissues. *Bioinformatics* 35, 3898–3905. <https://doi.org/10.1093/bioinformatics/btz196>.
12. King, A.N., Manning, C.F., and Trimmer, J.S. (2014). A Unique Ion Channel Clustering Domain on the Axon Initial Segment of Mammalian Neurons. *J Comp Neurol* 522, 2594–2608. <https://doi.org/10.1002/cne.23551>.
13. Bar, C., Kuchenbuch, M., Barcia, G., Schneider, A., Jennesson, M., Le Guyader, G., Lesca, G., Mignot, C., Montomoli, M., Parrini, E., et al. (2020). Developmental and epilepsy spectrum of KCNB1 encephalopathy with long-term outcome. *Epilepsia* 61, 2461–2473. <https://doi.org/10.1111/epi.16679>.
14. Coorens, T.H.H., Guillaumet-Adkins, A., Kovner, R., Linn, R.L., Roberts, V.H.J., Sule, A., and Van Hoose, P.M. (2025). The human and non-human primate developmental GTEx projects. *Nature* 637,

- 557–564. <https://doi.org/10.1038/s41586-024-08244-9>.
15. Leek, J.T., Johnson, W.E., Parker, H.S., Jaffe, A.E., Storey, J.D., and Kelso, J. (2012). The sva package for removing batch effects and other unwanted variation in high-throughput experiments. *BIOINFORMATICS APPLICATIONS NOTE* *28*, 882–883. <https://doi.org/10.1093/bioinformatics/bts034>.
 16. Lai, T., Jabaudon, D., Molyneaux, B.J., Azim, E., Arlotta, P., Menezes, J.R.L., and Macklis, J.D. (2008). SOX5 Controls the Sequential Generation of Distinct Corticofugal Neuron Subtypes. *Neuron* *57*, 232–247. <https://doi.org/10.1016/j.neuron.2007.12.023>.
 17. Joly-Amado, A., Kulkarni, N., and Nash, K.R. (2023). Reelin Signaling in Neurodevelopmental Disorders and Neurodegenerative Diseases. *Brain Sci* *13*, 1479. <https://doi.org/10.3390/brainsci13101479>.
 18. O’Brien, H.E., Hannon, E., Hill, M.J., Toste, C.C., Robertson, M.J., Morgan, J.E., McLaughlin, G., Lewis, C.M., Schalkwyk, L.C., Hall, L.S., et al. (2018). Expression quantitative trait loci in the developing human brain and their enrichment in neuropsychiatric disorders. *Genome Biology* *19*, 194. <https://doi.org/10.1186/s13059-018-1567-1>.
-